# Enhancing geometric representations for molecules with equivariant vector-scalar interactive message passing

Yusong Wang[1,2,6], Tong Wang [1,6] ✉, Shaoning Li[1,6], Xinheng He[1,3,4], Mingyu Li[1,5], Zun Wang [1], Nanning Zheng[2], Bin Shao [1] ✉ & Tie-Yan Liu [1]

Geometric deep learning has been revolutionizing the molecular modeling field. Despite the state-of-the-art neural network models are approaching ab initio accuracy for molecular property prediction, their applications, such as drug discovery and molecular dynamics (MD) simulation, have been hindered by insufficient utilization of geometric information and high computational costs. Here we propose an equivariant geometry-enhanced graph neural network called ViSNet, which elegantly extracts geometric features and efficiently models molecular structures with low computational costs. Our proposed ViSNet outperforms state-of-the-art approaches on multiple MD benchmarks, including MD17, revised MD17 and MD22, and achieves excellent chemical property prediction on QM9 and Molecule3D datasets. Furthermore, through a series of simulations and case studies, ViSNet can efficiently explore the conformational space and provide reasonable interpretability to map geometric representations to molecular structures.

Molecular modeling plays a crucial role in modern scientific and engineering fields, aiding in the understanding of chemical reactions, facilitating new drug development, and driving scientific and technological advancements[1–4]. One commonly used method in molecular modeling is density functional theory (DFT). DFT enables accurate calculations of energy, forces, and other chemical properties of molecules[5,6]. However, due to the large computational requirements, DFT calculations often demand significant computational resources and time, particularly for large molecular systems or high-precision calculations. Machine learning (ML) offers an alternative solution by learning from reference data with ab initio accuracy and high computational efficiency[7,8]. Gradient-domain machine learning (GDML)[9] constructs accurate molecular force fields using conservation of energy and limited samples from ab initio molecular dynamics trajectories, enabling cost-effective simulations while maintaining

accuracy. Symmetric GDML (sGDML)[10] further improves force field construction by incorporating physical symmetries, achieving CCSD(T)-level accuracy for flexible molecules. An exact iterative approach (Global sGDML)[11] extends sGDML to global force fields for molecules with several hundred atoms, maintaining correlations of atomic degree and accurately describing complex molecules and materials. In recent years, deep learning (DL) has demonstrated its powerful ability to learn from raw data without any hand-crafted features in many fields and thus attracted more and more attention. However, the inherent drawback of deep learning, which requires large amounts of data, has become a bottleneck for its application to more scenarios[12]. To alleviate the dependency on data for DL potentials, recent works have incorporated the inductive bias of symmetry into neural network design, known as geometric deep learning (GDL). Symmetry describes the conservation of physical laws, i.e., the

[1]Microsoft Research AI4Science, 100080 Beijing, China. [2]National Key Laboratory of Human–Machine Hybrid Augmented Intelligence, National Engineering Research Center for Visual Information and Applications, and Institute of Artificial Intelligence and Robotics, Xi'an Jiaotong University, 710049 Xi'an, China. [3]The CAS Key Laboratory of Receptor Research and State Key Laboratory of Drug Research, Shanghai Institute of Materia Medica, Chinese Academy of Sciences, 201203 Shanghai, China. [4]University of Chinese Academy of Sciences, 100049 Beijing, China. [5]Medicinal Chemistry and Bioinformatics Center, School of Medicine, Shanghai Jiaotong University, Shanghai 200025, China. [6]These authors contributed equally: Yusong Wang, Tong Wang, Shaoning Li. ✉e-mail: watong@microsoft.com; binshao@microsoft.com

unchanged physical properties with any transformations such as translations or rotations. It allows GDL to be extended to limited data scenarios without any data augmentation.

Equivariant graph neural network (EGNN) is one of the representative approaches in GDL, which has extensive capability to model molecular geometry[12–21]. A popular kind of EGNN conducts equivariance from directional information and involves geometric features to predict molecular properties. GemNet[20] extends the invariant Dime-Net/DimeNet++[16,17] with dihedral information. They explicitly extract geometric information in the Euclidean space with first-order geometric tensor, i.e., setting $l_{max} = 1$. PaiNN[18] and equivariant transformer[19] further adopt vector embedding and scalarize the angular representation implicitly via the inner product of the vector embedding itself. They reduce the complexity of explicit geometry extraction by taking the angular information into consideration. Another mainstream approach to achieving equivariance is through group representation theory, which can achieve higher accuracy but comes with large computational costs. NequIP, Allegro, and MACE[12,22,23] achieve state-of-the-art performance on several molecular dynamics simulation datasets leveraging high-order geometric tensors. On the one hand, algorithms based on group representation theory have strong mathematical foundations and are able to fully utilize geometric information using high-order geometric tensors. On the other hand, these algorithms often require computationally expensive operations such as the Clebsch–Gordan product (CG-product)[24], making them possibly suitable for periodic systems with elaborate model design but impractical for large molecular systems such as chemical and biological molecules without periodic boundary conditions.

In this study, we propose ViSNet (short for "Vector-Scalar interactive graph neural Network"), which alleviates the dilemma between computational costs and sufficient utilization of geometric information. By incorporating an elaborate runtime geometry calculation (RGC) strategy, ViSNet implicitly extracts various geometric features, i.e., angles, dihedral torsion angles, and improper angles in accordance with the force field of classical MD with linear time complexity, thus significantly accelerating model training and inference while reducing the memory consumption. To extend the vector representation, we introduce spherical harmonics and simplify the computationally expensive Clebsch–Gordan product with the inner product. Furthermore, we present a well-designed vector–scalar interactive equivariant message passing (ViS-MP) mechanism, which fully utilizes the geometric features by interacting vector hidden representations with scalar ones. When comprehensively evaluated on some benchmark datasets, ViSNet outperforms all state-of-the-art algorithms on all molecules in MD17, revised MD17 and MD22 datasets and shows superior performance on QM9, Molecule3D dataset indicating the powerful capability of molecular geometric representation. ViSNet also has won the PCQM4Mv2 track in the OGB-LCS@NeurIPS2022 competition (https://ogb.stanford.edu/neurips2022/results/). We then performed molecular dynamics simulations for each molecule on MD17 driven by ViSNet trained only with limited data (950 samples). The highly consistent interatomic distance distributions and the explored potential energy surfaces between ViSNet and quantum simulation illustrate that ViSNet is genuinely data-efficient and can perform simulations with high fidelity. To further explore the usefulness of ViSNet to real-world applications, we used an in-house dataset that consists of about 10,000 different conformations of the 166-atom mini-protein Chignolin derived from replica exchange molecular dynamics and calculated at the DFT level. When evaluated on the dataset, ViSNet also achieved significantly better performance than empirical force fields, and the simulations performed by ViSNet exhibited very close force calculation to DFT. In addition, ViSNet exhibits reasonable interpretability to map geometric representation to molecular

structures. The contributions of ViSNet can be summarized as follows:

- Proposing an RGC module that utilizes high-order geometric tensors to implicitly extract various geometric features, including angles, dihedral torsion angles, and improper angles, with linear time complexity.
- Introducing ViS-MP mechanism to enable efficient interaction between vector hidden representations and scalar ones and fully exploit the geometric information.
- Achieving state-of-the-art performance in six benchmarks for predicting energy, forces, HOMO-LUMO gap, and other quantum properties of molecules.
- Performing molecular dynamics simulations driven by ViSNet on both small molecules and 166-atom Chignolin with high fidelity.
- Demonstrating reasonable model interpretability between geometric features and molecular structures.

## Results
### Overview of ViSNet

ViSNet is a versatile EGNN that predicts potential energy, atomic forces as well as various quantum chemical properties by taking atomic coordinates and numbers as inputs. As shown in Fig. 1a, the model is composed of an embedding block and multiple stacked ViSNet blocks, followed by an output block. The atomic number and coordinates are fed into the embedding block followed by ViSNet blocks to extract and encode geometric representations. The geometric representations are then used to predict molecular properties through the output block. It is worth noting that ViSNet is an energy-conserving potential, i.e., the predicted atomic forces are derived from the negative gradients of the potential energy with respect to the coordinates[9,10].

The success of classical force fields shows that geometric features such as interatomic distances, angles, dihedral torsion angles, and improper angles in Fig. 2 are essential to determine the total potential energy of molecules. The explicit extraction of invariant geometric representations in previous studies often suffers from a large amount of time or memory consumption during model training and inference. Given an atom, the calculation of angular information scales $\mathcal{O}(\mathcal{N}^2)$ with the number of neighboring atoms, while the computational complexity is even $\mathcal{O}(\mathcal{N}^3)$ for dihedrals[20]. To alleviate this problem, inspired by Schütt et al.[18], we propose runtime geometry calculation (RGC), which uses an equivariant vector representation (termed as direction unit) for each node to preserve its geometric information. RGC directly calculates the geometric information from the direction unit which only sums the vectors from the target node to its neighbors once. Therefore, the computational complexity can be reduced to $\mathcal{O}(\mathcal{N})$. Notably, beyond employing angular information that has been used in PaiNN[18] and ET[19], ViSNet further considers the dihedral torsion and improper angle calculation with higher geometric tensors.

Considering the sub-structure of a toy molecule with four atoms shown in Fig. 2, the angular information of the target node $i$ could be conducted from the vector $\overrightarrow{r}_{ij}$ as follows:

$$\overrightarrow{u}_{ij} = \frac{\overrightarrow{r}_{ij}}{\left|\overrightarrow{r}_{ij}\right|}, \quad \overrightarrow{v}_i = \sum_{j=1}^{N_i} \overrightarrow{u}_{ij} \tag{1}$$

$$\left|\overrightarrow{v}_i\right|^2 = \sum_{j=1}^{N_i}\sum_{k=1}^{N_i} \left\langle \overrightarrow{u}_{ij}, \overrightarrow{u}_{ik} \right\rangle = \sum_{j=1}^{N_i}\sum_{k=1}^{N_i} \cos\theta_{jik} \tag{2}$$

where $\overrightarrow{r}_{ij}$ is the vector from node $i$ to its neighboring node $j$, $\overrightarrow{u}_{ij}$ is the unit vector of $\overrightarrow{r}_{ij}$. Here, we define the direction unit $\overrightarrow{v}_i$ as the sum of all unit vectors from node $i$ to its all neighboring nodes $j$, where node $i$ is the intersection of all unit vectors. As shown in Eq. (2), we calculate the inner product of the direction unit $\overrightarrow{v}_i$ which represents the sum of

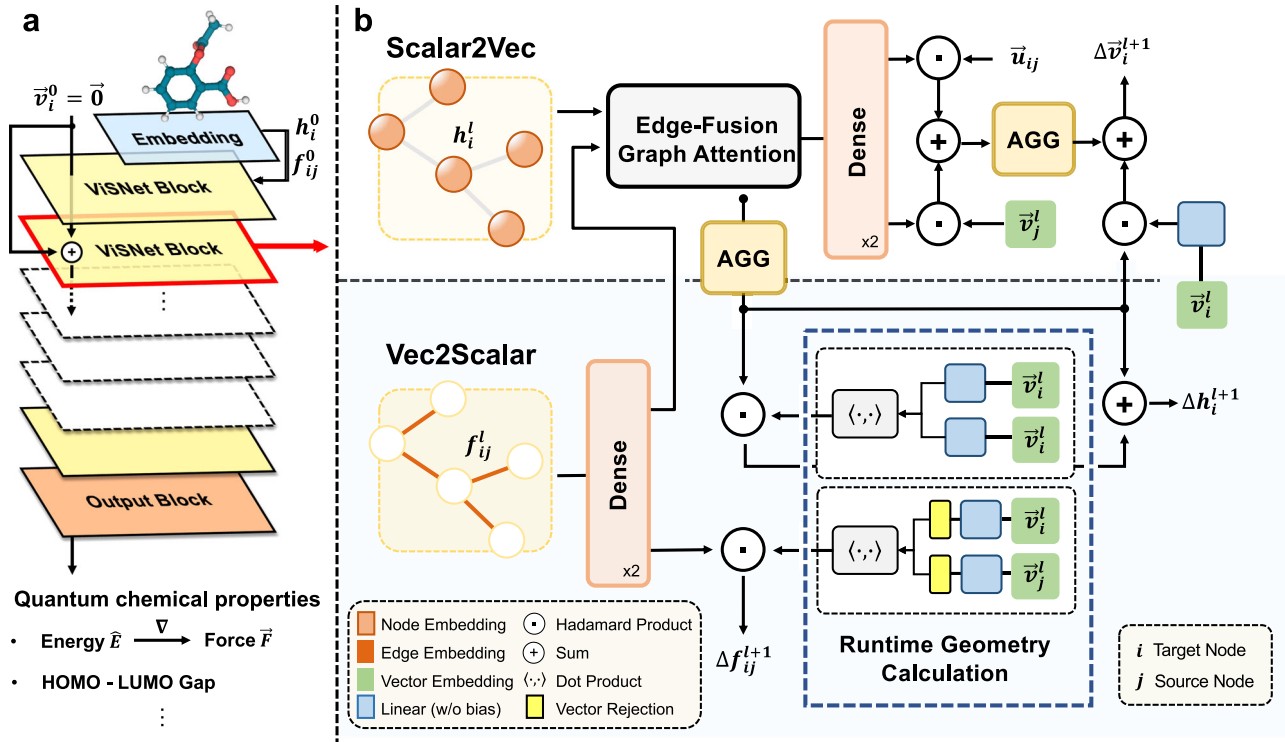

**Fig. 1 | The overall architecture of ViSNet. a** Model sketch of ViSNet. ViSNet embeds the 3D structures of molecules and extracts the geometric information through a series of ViSNet blocks and outputs the molecule properties such as energy, forces, and HOMO−LUMO gap through an output block. **b** Flowchart of one ViSNet Block. One ViSNet block consists of two modules: (i) *Scalar2Vec*, responsible for attaching scalar embeddings to vectors.; (ii) *Vec2Scalar*, renovates scalar embeddings built on RGC strategy. The inputs of Scalar2Vec are the node embedding $h_i$, edge embedding $f_{ij}$, direction unit $\vec{v}_i$ and the relative positions between two atoms. The edge-fusion graph attention module (serves as $\phi_m^s$) takes

as input $h_i$ and the output of the dense layer following $f_{ij}$, and outputs scalar messages. Before aggregation, each scalar message is transformed through a dense layer, and then fused with the unit of the relative position $\vec{u}_{ij}$ and its own direction unit $\vec{v}_j$. We further compute the vector messages and aggregate them all among the neighborhood. Through a gated residual connection, the final residual $\Delta\vec{v}_i$ is produced. In Vec2Scalar module, by Hadamard production of aggregated scalar messages and the output of RGC-Angle calculation and adding a gated residual connection, the final $\Delta h_i$ is figured out. Likewise, combining the projected $f_{ij}$ and the output of RGC-Dihedral calculation, the final $\Delta f_{ij}$ is determined.

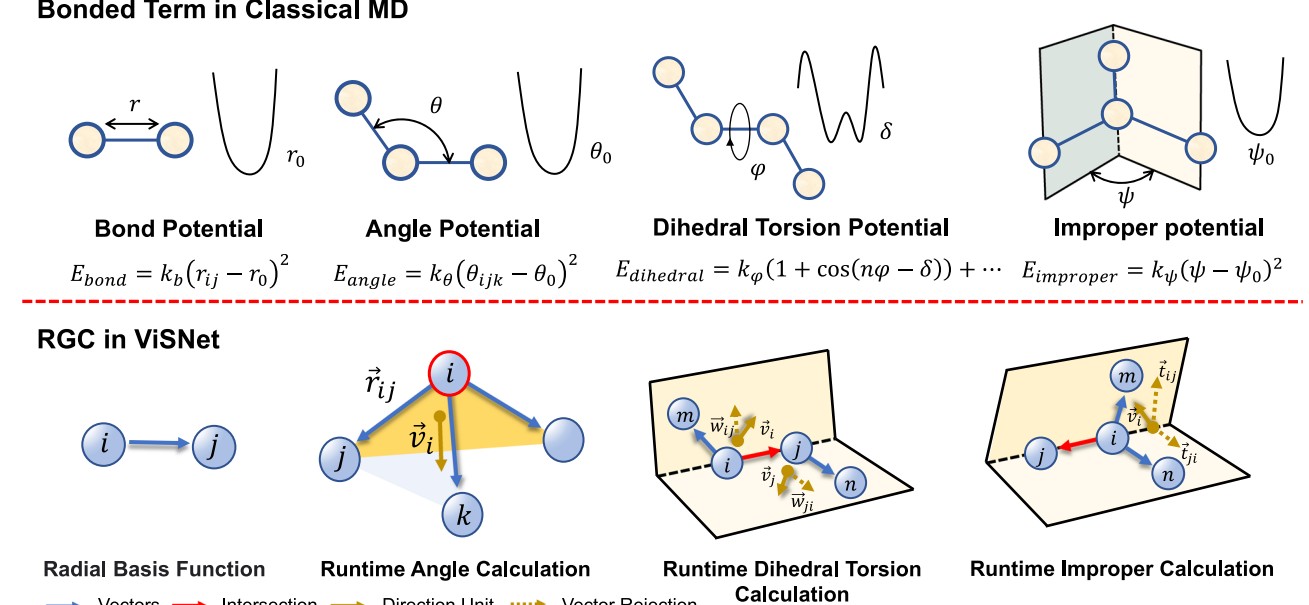

**Fig. 2 | Illustration of runtime geometry calculation (RGC) module and its relevance to the potential of bonded terms in classical molecular dynamics.** The bonded terms consist of bond length, bond angle, dihedral torsion, and

improper angle. The RGC module depicts all bonded terms of classical MD as model operations in linear time complexity. Yellow arrow $\vec{v}_i$ denotes the direction unit in Eq. (1).

the inner products of unit vectors from node $i$ to all its neighboring nodes. Combining with Eq. (1), the inner product of direction $\vec{v}_i$ finally stands for the sum of cosine values of all angles formed by node $i$ and any two of its neighboring nodes.

Similar to runtime angle calculation, we also calculate the vector rejection[25] of the direction unit $\vec{v}_i$ of node $i$ and $\vec{v}_j$ of node $j$ on the vector $\vec{u}_{ij}$ and $\vec{u}_{ji}$, respectively.

$$\vec{w}_{ij} = \text{Rej}_{\vec{u}_{ij}}\left(\vec{v}_i\right) = \vec{v}_i - \left\langle \vec{v}_i, \vec{u}_{ij}\right\rangle \vec{u}_{ij}$$
$$= \sum_{m=1}^{N_i} \text{Rej}_{\vec{u}_{ij}}\left(\vec{u}_{im}\right)$$
$$\vec{w}_{ji} = \text{Rej}_{\vec{u}_{ji}}\left(\vec{v}_j\right) = \vec{v}_j - \left\langle \vec{v}_j, \vec{u}_{ji}\right\rangle \vec{u}_{ji} \quad (3)$$
$$= \sum_{n=1}^{N_j} \text{Rej}_{\vec{u}_{ji}}\left(\vec{u}_{jn}\right)$$

where $\text{Rej}_{\vec{b}}(\vec{a})$ represents the vector component of $\vec{a}$ perpendicular to $\vec{b}$, termed as the vector rejection. $\vec{u}_{ij}$ and $\vec{v}_i$ are defined in Eq. (1). $\vec{w}_{ij}$ represents the sum of the vector rejection $\text{Rej}_{\vec{u}_{ij}}(\vec{u}_{im})$ and $\vec{w}_{ji}$ represents the sum of the vector rejection $\text{Rej}_{\vec{u}_{ji}}(\vec{u}_{jn})$. The inner product between $\vec{w}_{ij}$ and $\vec{w}_{ji}$ is then calculated to conduct dihedral torsion angle information of the intersecting edge $e_{ij}$ as follows:

$$\left\langle \vec{w}_{ij}, \vec{w}_{ji}\right\rangle = \sum_{m=1}^{N_i}\sum_{n=1}^{N_j}\left\langle \text{Rej}_{\vec{u}_{ij}}\left(\vec{u}_{im}\right), \text{Rej}_{\vec{u}_{ji}}\left(\vec{u}_{jn}\right)\right\rangle$$
$$= \sum_{m=1}^{N_i}\sum_{n=1}^{N_j} \cos\varphi_{mijn} \quad (4)$$

The improper angle is derived from a pyramid structure forming by 4 nodes. As the last toy molecule shown in Fig. 2, node $i$ is the vertex of the pyramid, and the improper torsion angle is formed by two adjacent planes with an intersecting edge $e_{ij}$. We can also calculate the improper angle by vector rejection:

$$\vec{t}_{ij} = \text{Rej}_{\vec{u}_{ij}}\left(\vec{v}_i\right) = \sum_{m=1}^{N_i} \text{Rej}_{\vec{u}_{ij}}\left(\vec{u}_{im}\right)$$
$$\vec{t}_{ji} = \text{Rej}_{\vec{u}_{ji}}\left(\vec{v}_i\right) = \sum_{n=1}^{N_i} \text{Rej}_{\vec{u}_{ji}}\left(\vec{u}_{in}\right) \quad (5)$$

In the same way, the inner product between $\vec{t}_{ij}$ and $\vec{t}_{ji}$ indicates the summation of improper angle information formed by $e_{ij}$:

$$\left\langle \vec{t}_{ij}, \vec{t}_{ji}\right\rangle = \sum_{m=1}^{N_i}\sum_{n=1}^{N_i}\left\langle \text{Rej}_{\vec{u}_{ij}}\left(\vec{u}_{im}\right), \text{Rej}_{\vec{u}_{ji}}\left(\vec{u}_{in}\right)\right\rangle$$
$$= \sum_{m=1}^{N_i}\sum_{n=1}^{N_i} \cos\psi_{mijn} \quad (6)$$

Multiple works have shown the effectiveness of high-order geometric tensors for molecular modeling[12,22,26,27]. However, the computational overheads of these approaches are generally expansive due to the CG-product, impeding their further application for large systems. In this work, we convert the vectors to high-order representation with spherical harmonics but discard CG-product with the inner product following the idea of RGC. We find that the extended high-order geometric tensors can still represent the above angular information in the

form of Legendre polynomials according to the addition theorem:

$$P_l(\cos\theta_{jik}) = P_l\left(\vec{u}_{ij} \cdot \vec{u}_{ik}\right)$$
$$\propto \sum_{m=-l}^{l} Y_{l,m}\left(\vec{u}_{ij}\right) Y_{l,m}^*\left(\vec{u}_{ik}\right) \quad (7)$$

where the $P_l$ is the Legendre polynomial of degree $l$, $Y_{l,m}$ denotes the spherical harmonics function and $Y_{l,m}^*$ denotes its complex conjugation. We sum the product of different order $l$ to obtain the scalar angular representation, which is the same operation as the inner product. It is worth noting that such an extension does not increase the model size and keeps the model architecture unchanged. We also provide proof about the rotational invariance of the RGC strategy in the section "Proofs of the rotational invariance of RGC".

In order to make full use of geometric information and enhance the interaction between scalars and vectors, we designed an effective vector–scalar interactive message-passing mechanism with respect to the intersecting nodes and edges for angles and dihedrals, respectively. It is important to note that previous studies[18,19] primarily focused on updating node features, whereas our approach updates both node and edge features during message passing, leading to a more comprehensive geometric representation. The key operations in ViS-MP are given as follows:

$$m_i^l = \sum_{j\in\mathcal{N}(i)} \phi_m^s\left(h_i^l, h_j^l, f_{ij}^l\right) \quad (8)$$

$$\vec{m}_i^l = \sum_{j\in\mathcal{N}(i)} \phi_m^v\left(m_{ij}^l, \vec{r}_{ij}, \vec{v}_j^l\right) \quad (9)$$

$$h_i^{l+1} = \phi_{un}^s\left(h_i^l, m_i^l, \left\langle \vec{v}_i^l, \vec{v}_i^l\right\rangle\right) \quad (10)$$

$$f_{ij}^{l+1} = \phi_{ue}^s\left(f_{ij}^l, \left\langle \text{Rej}_{\vec{r}_{ij}}\left(\vec{v}_i^l\right), \text{Rej}_{\vec{r}_{ji}}\left(\vec{v}_j^l\right)\right\rangle\right) \quad (11)$$

$$\vec{v}_i^{l+1} = \phi_{un}^v\left(\vec{v}_i^l, m_i^l, \vec{m}_i^l\right) \quad (12)$$

where $h_i$ denotes the scalar embedding of node $i$, $f_{ij}$ stands for the edge feature between node $i$ and node $j$. $\vec{v}_i$ represents the embedding of the direction unit mentioned in RGC. The superscript of variables indicates the index of the block that the variables belong to. We omit the improper angle here for brevity. A comprehensive version is depicted in Supplementary. ViS-MP extends the conventional message passing, aggregation, and update processes with vector–scalar interactions. Eqs. (8) and (9) depict our message-passing and aggregation processes. To be concrete, scalar messages $m_{ij}$ incorporating scalar embedding $h_j$, $h_i$, and $f_{ij}$ are passed and then aggregated to node $i$ through a message function $\phi_m^s$ (Eq. (8)). Similar operations are applied for vector messages $\vec{m}_i^l$ of node $i$ that incorporates scalar message $m_{ij}$, vector $\vec{r}_{ij}$ and vector embedding $\vec{v}_j$ (Eq. (9)). Equations (10) and (11) demonstrate the update processes. $h_i$ is updated by the aggregated scalar message output $m_i$ while the inner product of $\vec{v}_i$ is updated through an update function $\phi_{un}^s$. Then $\vec{f}_{ij}$ is updated by the inner product of the rejection of the vector embedding $\vec{v}_i$ and $\vec{v}_j$ through an update function $\phi_{ue}^s$. Finally, the vector embedding $\vec{v}_i$ is updated by both scalar and vector messages through an update function $\phi_{un}^v$. Notably, the vectors update function, i.e., $\phi^v$ require to be equivariant. The detailed message and update functions can be found in the Methods section. A proof about the equivariance of ViS-MP can be found in Supplementary Methods.

**Table 1 | Mean absolute errors (MAE) of energy (kcal/mol) and force (kcal/mol/Å) for 7 small organic molecules on MD17 compared with state-of-the-art algorithms**

| Molecule | | | SchNet | DimeNet | PaiNN | SpookyNet | ET | GemNet[a] | NequIP[b] | SO3KRATES | ViSNet |
|---|---|---|---|---|---|---|---|---|---|---|---|
| Aspirin | Energy | | 0.37 | 0.204 | 0.167 | 0.151 | 0.123 | – | 0.131 | 0.139 | **0.116** |
| | Forces | | 1.35 | 0.499 | 0.338 | 0.258 | 0.253 | 0.217 | 0.184 | 0.236 | **0.155** |
| Ethanol | Energy | | 0.08 | 0.064 | 0.064 | 0.052 | 0.052 | – | **0.051** | 0.061 | **0.051** |
| | Forces | | 0.39 | 0.230 | 0.224 | 0.094 | 0.109 | 0.085 | 0.071 | 0.096 | **0.060** |
| Malondialdehyde | Energy | | 0.13 | 0.104 | 0.091 | 0.079 | 0.077 | – | 0.076 | 0.077 | **0.075** |
| | Forces | | 0.66 | 0.383 | 0.319 | 0.167 | 0.169 | 0.155 | 0.129 | 0.147 | **0.100** |
| Naphthalene | Energy | | 0.16 | 0.122 | 0.116 | 0.116 | **0.085** | – | 0.113 | 0.115 | **0.085** |
| | Forces | | 0.58 | 0.215 | 0.077 | 0.089 | 0.061 | 0.051 | **0.039** | 0.074 | **0.039** |
| Salicylic acid | Energy | | 0.20 | 0.134 | 0.116 | 0.114 | 0.093 | – | 0.106 | 0.106 | **0.092** |
| | Forces | | 0.85 | 0.374 | 0.195 | 0.180 | 0.129 | 0.125 | 0.090 | 0.145 | **0.084** |
| Toluene | Energy | | 0.12 | 0.102 | 0.095 | 0.094 | **0.074** | – | 0.092 | 0.095 | **0.074** |
| | Forces | | 0.57 | 0.216 | 0.094 | 0.087 | 0.067 | 0.060 | 0.046 | 0.073 | **0.039** |
| Uracil | Energy | | 0.14 | 0.115 | 0.106 | 0.105 | **0.095** | – | 0.104 | 0.103 | **0.095** |
| | Forces | | 0.56 | 0.301 | 0.139 | 0.119 | 0.095 | 0.097 | 0.076 | 0.111 | **0.062** |

The best one in each category is highlighted in bold.
[a]The best results are reported among four variants of GemNet.
[b]NequIP only shows the results with $l = 3$.

In summary, the geometric features are extracted by inner products in the RGC strategy and the scalar and vector embeddings are cyclically updating each other in ViS-MP so as to learn a comprehensive geometric representation from molecular structures.

## Accurate quantum chemical property predictions

We evaluated ViSNet on several prevailing benchmark datasets including MD17[9,10,28], revised MD17[29], MD22[30], QM9[31], Molecule3D[32], and OGB-LSC PCQM4Mv2[33] for energy, force, and other molecular property prediction. MD17 consists of the MD trajectories of seven small organic molecules; the number of conformations in each molecule dataset ranges from 133,700 to 993,237. The dataset rMD17 is a reproduced version of MD17 with higher accuracy. MD22 is a recently proposed MD trajectories dataset that presents challenges with respect to larger system sizes (42–370 atoms). Large molecules such as proteins, lipids, carbohydrates, nucleic acids, and supramolecules are included in MD22. QM9 consists of 12 kinds of quantum chemical properties of 133,385 small organic molecules with up to 9 heavy atoms. Molecule3D is a recently proposed dataset including 3,899,647 molecules collected from PubChemQC with their ground-state structures and corresponding properties calculated by DFT. We focus on the prediction of the HOMO–LUMO gap following ComENet[34]. OGB-LSC PCQM4Mv2 is a quantum chemistry dataset originally curated under the PubChemQC including a DFT-calculated HOMO–LUMO gap of 3,746,619 molecules. The 3D conformations are provided for 3,378,606 training molecules but not for the validation and test sets. The training details of ViSNet on each benchmark are described in the "Methods" section.

We compared ViSNet with the state-of-the-art algorithms, including DimeNet[16], PaiNN[18], SpookyNet[21], ET[19], GemNet[20], UNiTE[35], NequIP[12], SO3KRATES[36], Allegro[22], MACE[23] and so on. As shown in Table 1 (MD17), Table 2 (rMD17), and Table 3 (MD22), it is remarkable that ViSNet outperformed the compared algorithms for both small (MD17 and rMD17) and large molecules (MD22) with the lowest mean absolute errors (MAE) of predicted energy and forces. On the one hand, compared with PaiNN, ET, and GemNet, ViSNet incorporated more geometric information and made full use of geometric information in ViS-MP, which contributes to the performance gains. On the other hand, compared with NequIP, Allegro, SO3KRATES, MACE, etc., ViSNet testified the effect of introducing spherical harmonics in the RGC module.

As shown in Table 4, ViSNet also achieved superior performance for chemical property predictions on QM9. It outperformed the compared algorithms for 9 of 12 chemical properties and achieved comparable results on the remaining properties. Elaborated evaluations on Molecule3D confirmed the high prediction accuracy of ViSNet as shown in Table 5. ViSNet achieved 33.6% and 6.51% improvements than the second-best for random split and scaffold split, respectively. Furthermore, ViSNet exhibited good portability to other multi-modality methods, e.g., Transformer-M[37] and outperformed other approaches on OGB-LSC PCQM4Mv2 (see Supplementary Fig. S1). ViSNet also achieved the winners of PCQM4Mv2 track in the OGB-LCS@NeurIPS2022 competition when testing on unseen molecules[38] (https://ogb.stanford.edu/neurips2022/results/).

To evaluate the computational efficiency of our ViSNet, following[23], we compare the time latency of ViSNet with prevailing models in Supplementary Fig. S2. The latency is defined as the time it takes to compute forces on a structure (i.e., the gradient calculation for a set of input coordinates through the whole deep neural network). As shown in Supplementary Fig. S2, ViSNet ($L = 2$) saved 42.8% time latency compared with MACE ($L = 2$). Notably, despite the use of CG-product, Allegro had a significant speed improvement compared to NequIP and BOTNet. However, ViSNet still saved 6.1%, 4.1%, and 61% time latency compared to Allegro with $L = 1$, 2, and 3, respectively.

## Efficient molecular dynamics simulations

To evaluate ViSNet as the potential for MD simulations, we incorporated ViSNet that trained only with 950 samples on MD17 into the ASE simulation framework[39] to perform MD simulations for all seven kinds of organic molecules. All simulations are run with a time step $\tau = 0.5$ fs under the Berendsen thermostat with the other settings the same as those of the MD17 dataset. As shown in Fig. 3, we analyzed the interatomic distance distributions derived from both AIMD simulations with ViSNet as the potential and ab initio molecular dynamics simulations at the DFT level for all seven molecules, respectively. As shown in Fig. 3a, the interatomic distance distribution $h(r)$ is defined as the ensemble average of atomic density at a radius $r^9$. Figure 3b–h illustrates the distributions derived from ViSNet are very close to those generated by DFT. We also compared the potential energy surfaces sampled by ViSNet and DFT for these molecules, respectively (Supplementary Fig. S3). The consistent potential energy surfaces suggest that ViSNet can recover the conformational space from the simulation

**Table 2 | Mean absolute errors (MAE) of energy (kcal/mol) and force (kcal/mol/Å) for 10 small organic molecules on rMD17 compared with state-of-the-art algorithms**

| Molecule | | UNiTE[a] | ACE | GemNet[b] | NequIP[b] | BOTNet | Allegro | MACE | ViSNet |
|---|---|---|---|---|---|---|---|---|---|
| Aspirin | Energy | 0.055 | 0.141 | – | 0.0530 | 0.0530 | 0.0530 | 0.0507 | **0.0445** |
| | Forces | 0.175 | 0.413 | 0.2191 | 0.1891 | 0.1960 | 0.1683 | 0.1522 | **0.1520** |
| Azobenzene | Energy | 0.025 | 0.083 | – | 0.0161 | 0.0161 | 0.0277 | 0.0277 | **0.0156** |
| | Forces | 0.097 | 0.251 | – | 0.0669 | 0.0761 | 0.0600 | 0.0692 | **0.0585** |
| Benzene | Energy | 0.002 | 0.0009 | – | 0.0009 | **0.0007** | 0.0069 | 0.0092 | **0.0007** |
| | Forces | 0.017 | 0.012 | 0.0115 | 0.0069 | 0.0069 | **0.0046** | 0.0069 | 0.0056 |
| Ethanol | Energy | 0.014 | 0.028 | – | 0.0092 | 0.0092 | 0.0092 | 0.0092 | **0.0078** |
| | Forces | 0.085 | 0.168 | 0.083 | 0.0646 | 0.0738 | **0.0484** | **0.0484** | 0.0522 |
| Malonaldehyde | Energy | 0.025 | 0.039 | – | 0.0184 | 0.0184 | 0.0138 | 0.0184 | **0.0132** |
| | Forces | 0.152 | 0.256 | 0.1522 | 0.1176 | 0.1338 | **0.0830** | 0.0945 | 0.0893 |
| Naphthalene | Energy | 0.011 | 0.021 | – | **0.0046** | **0.0046** | **0.0046** | 0.0115 | 0.0057 |
| | Forces | 0.060 | 0.118 | 0.0438 | 0.0300 | 0.0415 | **0.0208** | 0.0369 | 0.0291 |
| Paracetamol | Energy | 0.044 | 0.092 | – | 0.0323 | 0.0300 | 0.0346 | 0.0300 | **0.0258** |
| | Forces | 0.164 | 0.293 | – | 0.1361 | 0.1338 | 0.1130 | 0.1107 | **0.1029** |
| Salicylic acid | Energy | 0.017 | 0.042 | – | 0.0161 | 0.0184 | 0.0208 | 0.0208 | **0.0161** |
| | Forces | 0.088 | 0.214 | 0.1222 | 0.0922 | 0.0992 | **0.0669** | 0.0715 | 0.0795 |
| Toluene | Energy | 0.010 | 0.025 | – | 0.0069 | 0.0069 | 0.0092 | 0.0115 | **0.0059** |
| | Forces | 0.058 | 0.150 | 0.0507 | 0.0369 | 0.0438 | 0.0415 | 0.0346 | **0.0264** |
| Uracil | Energy | 0.013 | 0.025 | – | 0.0092 | 0.0092 | 0.0138 | 0.0115 | **0.0069** |
| | Forces | 0.088 | 0.152 | 0.0876 | 0.0715 | 0.0738 | **0.0415** | 0.0484 | 0.0495 |

The best one in each category is highlighted in bold.
[a]For a fair comparison, the "direct learning" results without any extra input are compared.
[b]The best results are reported among four variants of GemNet and four orders $l \in \{0, 1, 2, 3\}$ of NequIP.

**Table 3 | Mean absolute errors (MAE) of energy (kcal/mol) and force (kcal/mol/Å) for 7 large-scale molecules on MD22**

| Molecules | | Ac-Ala3-NHMe | AT-AT | AT-AT-CG-CG | DHA | Buckyball catcher | Stachyose | Double-walled nanotube |
|---|---|---|---|---|---|---|---|---|
| sGDML | Energy | 0.391 | 0.720 | 1.42 | 1.29 | 1.17 | 4.00 | 4.00 |
| | Forces | 0.790 | 0.690 | 0.700 | 0.750 | 0.680 | 0.680 | 0.520 |
| ViSNet | Energy | 0.0636 | 0.0708 | **0.196** | 0.0741 | **0.508** | 0.0915 | 0.800 |
| | Forces | 0.0830 | 0.0812 | 0.148 | 0.0598 | **0.184** | 0.0879 | 0.362 |
| ViSNet-improper | Energy | **0.0546** | **0.0668** | 0.197 | **0.0700** | 0.537 | **0.0882** | **0.601** |
| | Forces | **0.0709** | **0.0776** | **0.139** | **0.0554** | 0.201 | **0.0802** | **0.292** |

The number of training splits is the same as sGDML. ViSNet-improper contains the runtime improper calculation in the ViS-MP. The best one in each category is highlighted in bold.

**Table 4 | Mean absolute errors (MAE) of 12 kinds of molecular properties on QM9 compared with state-of-the-art algorithms**

| Target | Unit | SchNet | EGNN | DimeNet++ | PaiNN | SphereNet | PaxNet | ET | ComENet | ViSNet |
|---|---|---|---|---|---|---|---|---|---|---|
| $\mu$ | mD | 33 | 29 | 29.7 | 12 | 24.5 | 10.8 | 11 | 24.5 | **9.5** |
| $\alpha$ | $ma_0^3$ | 235 | 71 | 43.5 | 45 | 44.9 | 44.7 | 59 | 45.2 | **41.1** |
| $\epsilon_{HOMO}$ | meV | 41 | 29 | 24.6 | 27.6 | 22.8 | 22.8 | 20.3 | 23.1 | **17.3** |
| $\epsilon_{LUMO}$ | meV | 34 | 25 | 19.5 | 20.4 | 18.9 | 19.2 | 17.5 | 19.8 | **14.8** |
| $\Delta\epsilon$ | meV | 63 | 48 | 32.6 | 45.7 | 31.1 | **31** | 36.1 | 32.4 | 31.7 |
| $\langle R^2 \rangle$ | $ma_0^2$ | 73 | 106 | 331 | 66 | 268 | 93 | 33 | 259 | **29.8** |
| ZPVE | meV | 1.7 | 1.55 | 1.21 | 1.28 | **1.12** | 1.17 | 1.84 | 1.2 | 1.56 |
| $U_0$ | meV | 14 | 11 | 6.32 | 5.85 | 6.26 | 5.9 | 6.15 | 6.59 | **4.23** |
| $U$ | meV | 19 | 12 | 6.28 | 5.83 | 6.36 | 5.92 | 6.38 | 6.82 | **4.25** |
| $H$ | meV | 14 | 12 | 6.53 | 5.98 | 6.33 | 6.04 | 6.16 | 6.86 | **4.52** |
| $G$ | meV | 14 | 12 | 7.56 | 7.35 | 7.78 | 7.14 | 7.62 | 7.98 | **5.86** |
| $C_v$ | mcal/mol K | 33 | 31 | 23 | 24 | **22** | 23.1 | 26 | 24 | 23 |

The best one in each category is highlighted in bold.

trajectories. Moreover, compared to DFT, numerous groundbreaking machine learning force fields (MLFFs), including sGDML[10], ANI[40], DPMD[41], and PhysNet[42] have proven their exceptional speeds in MD simulations. Similar to such algorithms, ViSNet also exhibited significant computational cost reduction compared to DFT as shown in Supplementary Fig. S4 and Table S2.

To further examine the molecular properties derived from simulations driven by ViSNet, we performed 500 ps MD simulations at a constant energy ensemble (NVE) for ethanol in the MD17 dataset with a time step of $\tau = 0.5$ fs and 200 ps Ac-Ala3-NHMe in the MD22 dataset with a time step of $\tau = 1$ fs. The simulations were driven by ViSNet, sGDML, and DFT, respectively. For ethanol, we analyzed its vibrational spectra and the probability distribution of dihedral angles. For Ac-Ala3-NHMe, we investigated its vibrational spectra and potential energy surface (PES) via the Ramachandran plot. To analyze the Ramachandran plot of different simulations, the free energy value was estimated using the potential of mean force (PMF). $\phi$ and $\psi$ were set as two reaction coordinates $(x, y)$. All three $\phi$ and $\psi$ dihedrals in Ac-Ala3-NHMe were calculated and plotted. The relative free energy value was calculated and referred to with the minimum value. To generate the landscape, 40 bins were used in both the $x$ and $y$ directions. Supplementary Fig. S5a and b demonstrate that both ViSNet and sGDML generate similar vibrational spectra, with slight differences in peak

intensities compared to DFT. The probability distribution of hydroxyl angles in ethanol (Supplementary Fig. S5c) reveals three minima: gauche $\pm$ ($M_{g\pm}$) and *trans* ($M_t$). Furthermore, even though ViSNet showed better performance than sGDML for various conformations in the MD22 dataset, starting from the same structure of the alanine tetrapeptide, the performance difference may not have a notable impact on the sampling efficiency for such small molecules, and thus may also lead to similar dynamics on the Ramachandran plots as shown in the Supplementary Fig. S5d–f. These results demonstrate that with only a few training samples, ViSNet can act with the potential to perform high-fidelity molecular dynamics simulations with much less computational cost and higher accuracy.

### Applications for real-world full-atom proteins

To examine the usefulness of ViSNet in real-world applications, we made evaluations on the 166-atom mini-protein Chignolin (Fig. 4a). Based on a Chignolin dataset consisting of about 10,000 conformations that sampled by replica exchange MD[43] and calculated at DFT level by Gaussian 16[44] in our another study[45,46], we split it as training, validation, and test sets by the ratio of 8:1:1. We trained ViSNet as well as other prevailing MLFFs including ET[19], PaiNN[18], GemNet-OC[47], MACE[23], NequIP[12] and Allegro[22] and compared them with molecular mechanics (MM)[48]. The DFT results were used as the ground truth. Figure 4b shows the free energy landscape of Chignolin and is depicted by $d_{D3–G7}$ (the distance between carbonyl oxygen on the D3 backbone and nitrogen on the G7 backbone) and $d_{E5–T8}$ (the distance between carbonyl oxygen on the E5 backbone and nitrogen on T8 backbone). The concentrated energy basin on the left shows the folded state and the scattered energy basin on the right shows the unfolded state. We randomly selected six structures from different regions of the potential energy surface for visualization. Among them, four structures were predicted by the model with smaller errors than the MAE while the other two with larger errors. Interestingly, all models consistently performed poorly on the structures with high potential energies (low probability of sampling) and performed well on the other structures. This implies that the sampling of conformations with high potential energies could be enhanced to ensure the generalization ability of the models.

**Table 5 | Mean absolute errors (MAE) of HOMO–LUMO gap (eV) on Molecule3D test set for both random and scaffold splits compared with state-of-the-art algorithms**

| Model | Random | Scaffold |
|---|---|---|
| GIN-Virtual | 0.1036 | 0.2371 |
| SchNet | 0.0428 | 0.1511 |
| DimeNet++ | 0.0306 | 0.1214 |
| SphereNet | 0.0301 | 0.1182 |
| ComENet | 0.0326 | 0.1273 |
| ViSNet | **0.0200** | **0.1105** |

The best one in each category is highlighted in bold.

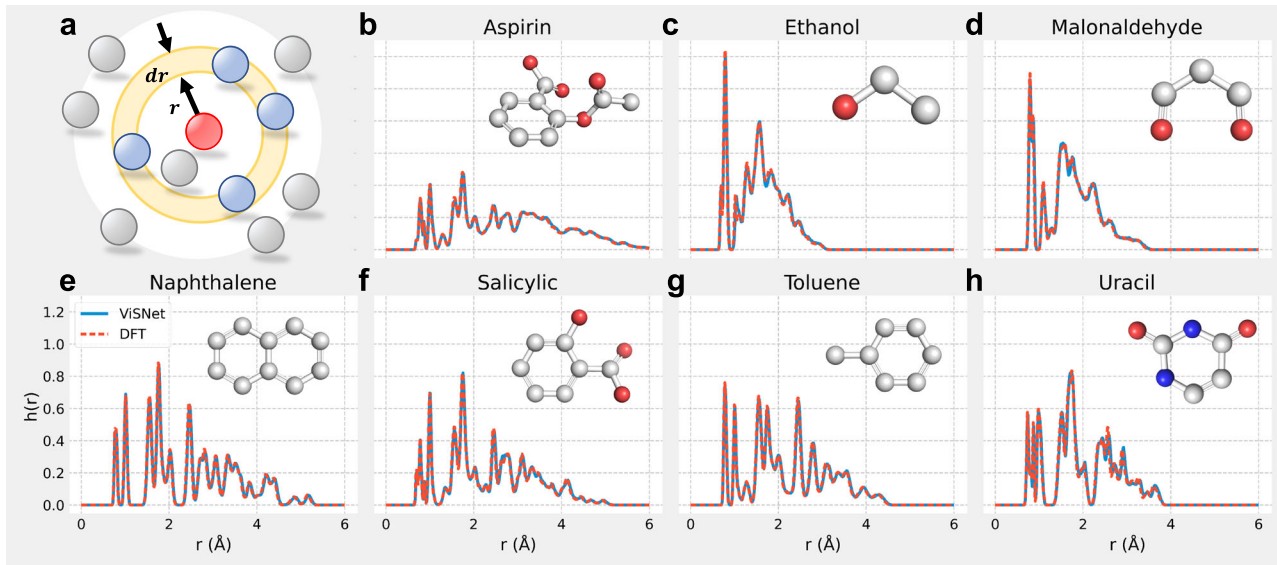

**Fig. 3 | The interatomic distance distributions of MD simulations driven by ViSNet and DFT. a** An illustration about the atomic density at a radius *r* with the arbitrary atom as the center. The interatomic distance distribution is defined as the ensemble average of atomic density. **b–h** The interatomic distance distributions comparison between simulations by ViSNet and DFT for all seven organic molecules in MD17. The curve of ViSNet is shown using a solid blue line, while the dashed orange line is used for the DFT curve. The structures of the corresponding molecules are shown in the upper right corner. Source data are provided as a Source Data file.

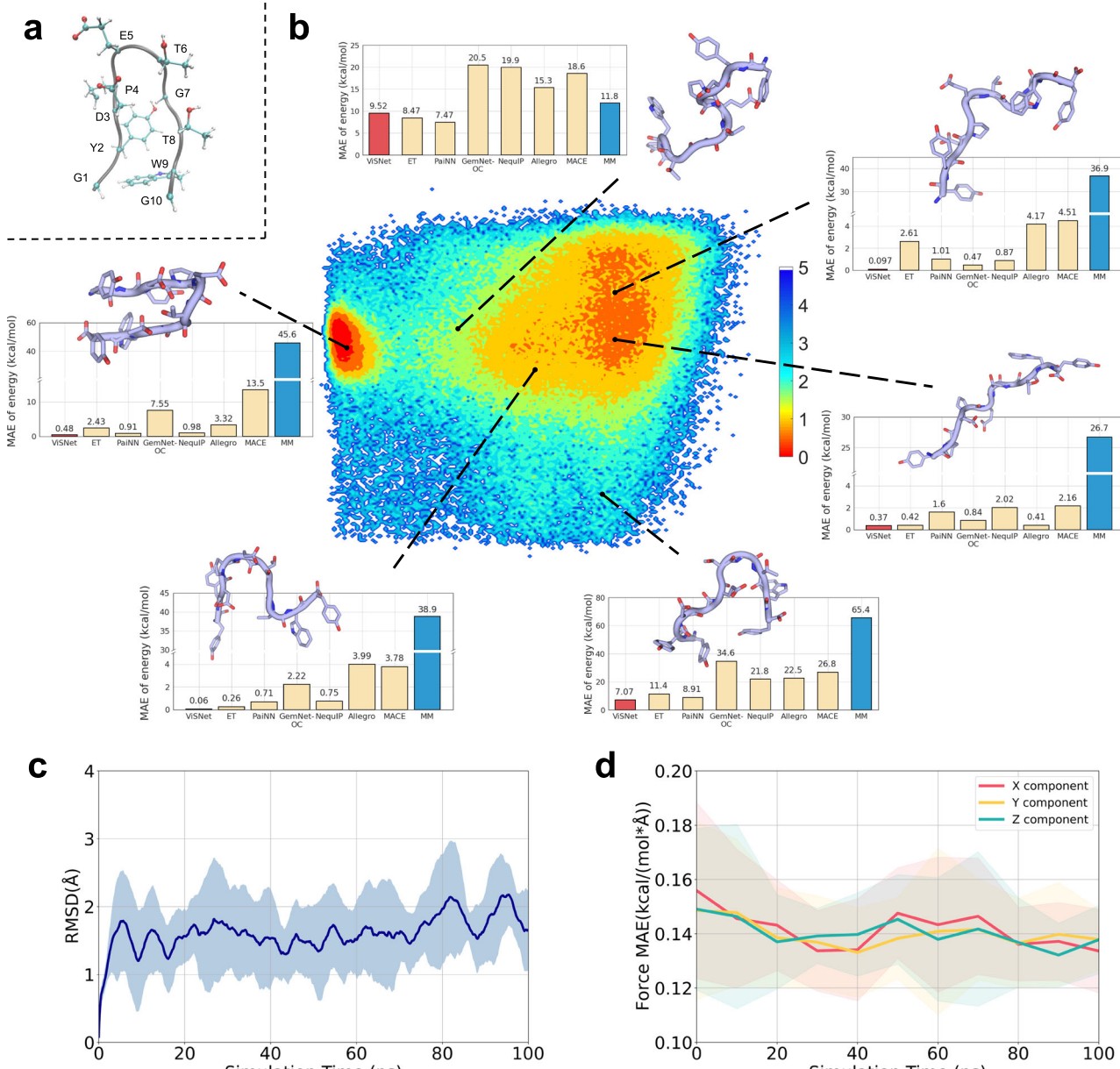

**Fig. 4 | Applications of ViSNet for Chignolin conformational space evaluation and MD simulations. a** The visualization of Chignolin structure. The backbone is colored grey while the side chains of each residue in Chignolin are highlighted with a ball and stick. **b** The energy landscape of Chignolin sampled by REMD. The *x*-axis of the landscape is the distance between carbonyl oxygen on the D3 backbone and nitrogen on the G7 backbone, while the *y*-axis is the distance between carbonyl oxygen on the E5 backbone and nitrogen on the T8 backbone. Six structures were then selected for visualization. Each structure is shown as a cartoon and residues are depicted in sticks. The histograms show the absolute error between the energy difference predicted by MLFFs including ViSNet, ET, PaiNN, GemNet-OC, NequIP, Allegro, and MACE or calculated by MM, and the ground truth calculated by DFT on the corresponding structure. **c** The average root mean square deviation (RMSD) of the Chignolin trajectories simulated by ViSNet was calculated from 10 different trajectories. The shaded areas indicate the standard deviation range. **d** The MAE of each component of atomic forces during the simulations driven by ViSNet. The ground truth energies and forces were calculated using Gaussian 16. The shaded areas indicate the standard deviation range. Source data are provided as a Source Data file.

Supplementary Fig. S6 shows the correlations between the energies predicted by MLFFs or MM and the ground truth values calculated by DFT for all conformations in the test set. ViSNet achieved a lower MAE and a higher $R^2$ score. From the violin plot of the absolute errors shown in Supplementary Fig. S7, ViSNet, PaiNN and ET exhibited smaller errors than other MLFFs while MM got a much wider range of prediction errors. Similar results can be seen in the force correlations in each component shown in Supplementary Fig. S8. Detailed settings about DFT and MM calculations are shown in Supplementary Materials. Furthermore, we also made a comprehensive comparison by taking model performance, training time consumption, and model size into consideration. ViSNet and other state-of-the-art algorithms such as PaiNN, ET, GemNet-OC, MACE, NequIP, and Allegro were analyzed on the Chignolin dataset and shown in Fig. 5. Although ViSNet is marginally slower than ET and PaiNN, it introduces more geometric information, significantly enhancing its performance. When compared to GemNet, which also incorporates dihedral angles, ViSNet's computational cost is significantly more affordable. Similarly, ViSNet proves to be computationally efficient when compared to models employing the CG-product method, such as MACE, Allegro, and NequIP.

In addition, we performed MD simulations for Chignolin driven by ViSNet. 10 conformations were randomly selected as initial structures,

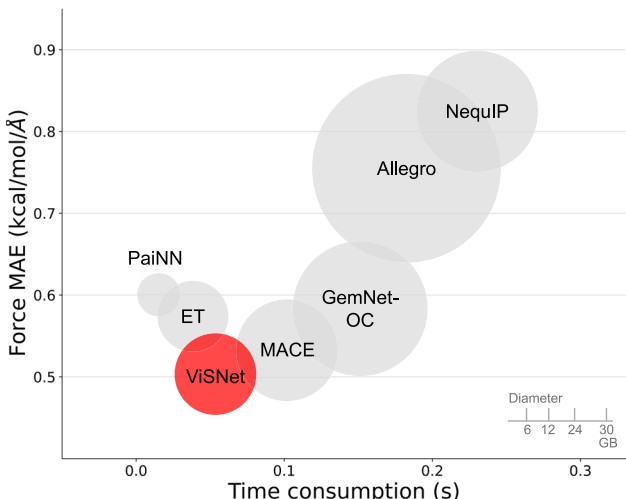

**Fig. 5 | The comparison of model performance (*y*-axis), training time consumption (*x*-axis), and training memory consumption (volume) among ViSNet (red) and other algorithms (grey) including PaiNN, ET, MACE, GemNet-OC, Allegro, and NequIP on Chignolin.** PaiNN and ET are faster and smaller as ViSNet further incorporates dihedral calculation. ViSNet outperforms GemNet-OC due to its Runtime Geometry Calculation, reducing the explicit extraction of dihedral complexity from $\mathcal{O}(\mathcal{N}^3)$ to $\mathcal{O}(\mathcal{N})$. Additionally, ViSNet is also faster and smaller than MACE, Allegro, and NequIP for streamlining the CG-product. ViSNet achieves the best performance for its elaborate design, i.e., runtime geometric calculation and vector–scalar interactive message passing. Source data are provided as a Source Data file.

and 100 ps simulations were run for each. As shown in Fig. 4c, the RMSD for 10 simulation trajectories is shown against the simulation time. In Fig. 4d, we displayed the MAE values of each component of the atomic forces between ViSNet and those calculated by Gaussian 16[44] at the DFT level. The simulation trajectory driven by ViSNet exhibited a small force difference for each component to quantum mechanics, which implies that ViSNet has no bias towards any force component, and thus consolidates the accuracy and potential usefulness for real-world applications.

### Interpretability of ViSNet on molecular structures

Prior works have shown the effectiveness of incorporating geometric features, such as angles[16,20]. The primary method of geometry extraction utilized by ViSNet is the distinct inner product in its runtime geometry calculation. To this end, we illustrate a reasonable model interpretability of ViSNet by mapping the angle representations derived from the inner product of direction units in the model to the atoms in the molecular structure. We aim to bridge the gap between geometric representation in ViSNet and molecular structures. We visualized the embeddings after the inner product of direction units $\langle \vec{v}_i, \vec{v}_i \rangle$ extracted from 50 aspirin samples on the validation set. The high-dimensional embeddings were reduced to 2-dimensional space using T-SNE[49] and then clustered using DBSCAN[50] without the prior of number of clusters.

Supplementary Fig. S9 exhibits the clustering results of nodes' embeddings after the inner product of their corresponding direction units. We further map the clustered nodes to the atoms of aspirin chemical structure. Interestingly, the embeddings for these nodes could be distinctly gathered into several clusters shown in different colors. For example, although carbon atom $C_{11}$ and carbon atom $C_{12}$ possess different positions and connect with different atoms, their inner product $\langle \vec{v}_i, \vec{v}_i \rangle$ are clustered into the same class for holding similar substructures ($\{C_{11}-O_2O_3C_6\}$ and $\{C_{12}-O_1O_4C_{13}\}$). To summarize, ViSNet can discriminate different molecular substructures in the embedding space.

### Ablation study

To further explore where the performance gains of ViSNet come from, we conducted a comprehensive ablation study. Specifically, we excluded the runtime angle calculation (w/o A), runtime dihedral calculation (w/o D), and both of them (w/o A&D) in ViSNet, in order to evaluate the usefulness of each part. ViSNet-improper denotes the additional improper angles and ViSNet$_{l=1}$ uses the first-order spherical harmonics.

We designed some model variants with different message-passing mechanisms based on ViS-MP for scalar and vector interaction. ViSNet-N directly aggregates the dihedral information to intersecting nodes, and ViSNet-T leverages another form of dihedral calculation. The details of these model variants are elaborated in Supplementary. The results of the ablation study are shown in Supplementary Table S3 and Supplementary Fig. S10. Based on the results, we can see that both kinds of directional geometric information are useful and the dihedral information contributes a little bit more to the final performance. The significant performance drop from ViSNet-N and ViSNet-T further validates the effectiveness of the ViS-MP mechanism. ViSNet-improper achieves similar performance to ViSNet for small molecules, but the contribution of improper angles is more obvious for large molecules (see Table 3). Furthermore, ViSNet using higher-order spherical harmonics achieves better performance.

## Discussion

We propose ViSNet, a geometric deep learning potential for molecular dynamics simulation. The group representation theory-based methods and the directional information-based methods are two mainstream classes of geometric deep learning potentials to enforce SE(3) equivariance[20]. ViSNet takes advantage of both sides in designing the RGC strategy and ViS-MP mechanism. On the one hand, the RGC strategy explicitly extracts and exploits the directional geometric information with computationally lightweight operations, making the model training and inference fast. On the other hand, ViS-MP employs a series of effective and efficient vector-scalar interactive operations, leading to the full use of geometric information. Furthermore, according to the many-body expansion theory[51–53], the potential energy of the whole system equals the potential of each single atom plus the energy corrections from two-bodies to many-bodies. Most of the previous studies model the truncated energy correction terms hierarchically with *k*-hop information via stacking *k* message passing blocks. Different from these approaches, ViSNet encodes the angle, dihedral torsion, and improper information in a single block, which empowers the model to have a much more powerful representation ability. In addition, ViSNet's universality or completeness is not validated by the geometric Weisfeiler–Leman (GWL) test[54] due to the inner product operation, which is computationally efficient but fails to distinguish certain atom reflection structures with the same angular information. To pass counterexamples or the GWL test, incorporating the CG-product with higher-order spherical harmonics is necessary in future studies.

Besides predicting energy, force, and chemical properties with high accuracy, performing molecular dynamics simulations with ab initio accuracy at the cost of the empirical force field is a grand challenge. ViSNet proves its usefulness in real-world ab initio molecular dynamics simulations with less computational costs and the ability of scaling to large molecules such as proteins. Extending ViSNet to support larger and more complex molecular systems will be our future research direction.

## Methods

### Equivariance

In the context of machine learning for atomic systems, equivariance is a pervasive concept. Specifically, the atomic vectors such as dipoles or forces must rotate in a manner consistent with the conformation

coordinates. In molecular dynamics, such equivariance can be ensured by computing gradients based on a predicted conservative scalar energy. Formally, a function $\mathcal{F}: \mathcal{X} \rightarrow \mathcal{Y}$ is equivariant should guarantee:

$$\mathcal{F}(\rho_{\mathcal{X}}(g) \circ x) = \rho_{\mathcal{Y}}(g) \circ \mathcal{F}(x), \qquad (13)$$

where $\rho_{\mathcal{X}}(g)$ and $\rho_{\mathcal{X}}(g)$ are group representations in input and output spaces. The integration of equivariance into model parameterization has been shown to be effective, as seen in the implementation of shift-equivariance in CNNs, which is critical for enhancing the generalization capacity.

## Proofs of the rotational invariance of RGC
Assume that the molecule rotates in 3D space, i.e.,

$$\vec{r}'_{ij} = R\vec{r}_{ij} \qquad (14)$$

where, $R \in SO(3)$ is an arbitrary rotation matrix that satisfies:

$$\det |R| = 1, R^{\mathrm{T}} R = I \qquad (15)$$

The angular information after rotation is calculated as follows:

$$\vec{u}'_{ij} = \frac{\vec{r}'_{ij}}{|\vec{r}'_{ij}|} = \frac{R\vec{r}_{ij}}{\det |R| \cdot |\vec{r}_{ij}|} = R\vec{u}_{ij} \qquad (16)$$

$$\vec{v}'_i = \sum_{j=1}^{N_i} \vec{u}'_{ij} = R \sum_{j=1}^{N_i} \vec{u}_{ij} = R\vec{v}_i \qquad (17)$$

$$\begin{aligned} \left|\vec{v}'_i\right|^2 &= \left\langle \vec{v}'_i, \vec{v}'_i \right\rangle = \left(\vec{v}'_i\right)^{\mathrm{T}} \vec{v}'_i \\ &= \vec{v}_i^{\mathrm{T}} R^{\mathrm{T}} R \vec{v}_i = \left\langle \vec{v}_i, \vec{v}_i \right\rangle = \left|\vec{v}_i\right|^2 \end{aligned} \qquad (18)$$

As shown in Eq. (18), the angle information does not change after rotation. The dihedral angular and improper information is also rotationally invariant since:

$$\vec{w}'_{ij} = \vec{v}'_i - \left\langle \vec{v}'_i, \vec{u}'_{ij} \right\rangle \vec{u}'_{ij} = R\vec{v}_i - \left\langle R\vec{v}_i, R\vec{u}_{ij} \right\rangle R\vec{u}_{ij} \qquad (19)$$

As Eq. (18) proved, the inner product has rotational invariance. Then, Eq. (19) can be further simplified as

$$\vec{w}'_{ij} = R\left(\vec{v}_i - \left\langle \vec{v}_i, \vec{u}_{ij} \right\rangle \vec{u}_{ij}\right) = R\vec{w}_{ij} \qquad (20)$$

The dihedral or improper angular information after rotation is calculated as:

$$\left\langle \vec{w}'_{ij}, \vec{w}'_{ji} \right\rangle = \left\langle R\vec{w}_{ij}, R\vec{w}_{ji} \right\rangle = \left\langle \vec{w}_{ij}, \vec{w}_{ji} \right\rangle \qquad (21)$$

As a result, Eqs. (18) and (21) have proved the rotational invariance of our proposed runtime geometry calculation (RGC).

We also provide proof of the equivariance of our ViS-MP in Supplementary Methods.

## Detailed operations and modules in ViSNet
ViSNet predicts the molecular properties (e.g., energy $\hat{E}$, forces $\vec{F} \in \mathbb{R}^{N \times 3}$, dipole moment $\mu$) from the current states of atoms, including the atomic positions $X \in \mathbb{R}^{N \times 3}$ and atomic numbers $Z \in \mathbb{N}^N$.

The architecture of the proposed ViSNet is shown in Fig. 1. The overall design of ViSNet follows the vector–scalar interactive message passing as illustrated from Eqs. (8)–(11). First, an embedding block encodes the atom numbers and edge distances into the embedding space. Then, a series of ViSNet blocks update the node-wise scalar and vector representations based on their interactions. A residual connection is placed between two ViSNet blocks. Finally, stacked corresponding gated equivariant blocks proposed by[18] are attached to the output block for specific molecular property prediction.

**The embedding block.** ViSNet expands the direct node and edge embedding with their neighbors. It first embeds atomic chemical symbol $z_i$, and calculates the edge representation whose distances within the cutoff through radial basis functions (RBF). Then the initial embedding of the atom $i$, its 1-hop neighbors $j$ and the directly connected edge $e_{ij}$ within cutoff are fused together as the initial node embedding $h_i^0$ and edge embedding $f_{ij}^0$. In summary, the embedding block is given by:

$$h_i^0, f_{ij}^0 = \text{Embedding Block}\left(z_i, z_j, e_{ij}\right), \quad j \in \mathcal{N}(i) \qquad (22)$$

$\mathcal{N}(i)$ denotes the set of 1-hop neighboring nodes of node $i$, and $j$ is one of its neighbors. The embedding process is elaborated in Supplementary. The initial vector embedding $\vec{v}_i$ is set to $\vec{0}$. The vector embeddings $\vec{v}$ are projected into the embedding space by following[18]; $\vec{v} \in \mathbb{R}^{N \times 3 \times F}$ and $F$ is the size of hidden dimension. The advantage of such projection is to assign a unique high-dimensional representation for each embedding to discriminate from each other. Further discussions on its effectiveness and interpretability are given in the Results section.

**The Scalar2Vec module.** In the Scalar2Vec module, the vector embedding $\vec{v}$ is updated by both the scalar messages derived from node and edge scalar embeddings (Eq. (8)) and the vector messages with inherent geometric information (Eq. (9)). The message of each atom is calculated through an Edge-Fusion Graph Attention module, which fuses the node and edge embeddings and computes the attention scores. The fusion of the node and edge embeddings could be the concatenation operation, Hadamard product, or adding a learnable bias[55]. We leverage the Hadamard product and the vanilla multi-head attention mechanism borrowed from Transformer[56] for edge-node fusion.

Following[19], we pass the fused representations through a nonlinear activation function as shown in Eq. (23). The value ($V$) in the attention mechanism is also fused by edge features before being multiplied by attention scores weighted by a cosine cutoff as shown in Eq. (24),

$$\alpha_{ij}^l = \sigma\left(\left(W_Q^l h_i^l\right)\left(W_K^l h_j^l \odot \text{Dense}_K^l\left(f_{ij}^l\right)\right)^{\mathrm{T}}\right) \qquad (23)$$

$$m_{ij}^l = \alpha_{ij}^l \cdot \phi\left(\left|\vec{r}_{ij}\right|\right) \cdot \left(W_V^l h_j^l \odot \text{Dense}_V^l\left(f_{ij}^l\right)\right) \qquad (24)$$

where $l \in \{0, 1, 2, \cdots, L\}$ is the index of the block, $\sigma$ denotes the activation function (SiLU in this paper), $W$ is the learnable weight matrix, $\odot$ represents the Hadamard product, $\phi(\cdot)$ denotes the cosine cutoff and Dense$(\cdot)$ refers to one learnable weight matrix with an activation function. For brevity, we omit the learnable bias for linear transformation on scalar embedding in equations, and there is no bias for vector embedding to ensure the equivariance.

Then, the computed $m_{ij}^l$ is used to produce the geometric messages $\vec{m}_{ij}^l$ for vectors:

$$\vec{m}_{ij}^l = \left(\text{Dense}_u^l\left(m_{ij}^l\right) \odot \vec{u}_{ij}\right) + \left(\text{Dense}_v^l\left(m_{ij}^l\right) \odot \vec{v}_j^l\right) \qquad (25)$$

And the vector embedding $\vec{v}^{\,l}$ is updated by:

$$m_i^l = \sum_{j \in \mathcal{N}(i)} m_{ij}^l, \quad \vec{m}_i^l = \sum_{j \in \mathcal{N}(i)} \vec{m}_{ij}^l \qquad (26)$$

$$\Delta \vec{v}_i^{\,l+1} = \vec{m}_i^l + W_{vm}^l m_i^l \odot W_v^l \vec{v}_i^{\,l} \qquad (27)$$

**The Vec2Scalar module.** In the Vec2Scalar module, the node embedding $h_i^l$ and edge embedding $f_{ij}^l$ are updated by the geometric information extracted by the RGC strategy, i.e., angles (Eq. (10)) and dihedrals (Eq. (11)), respectively. The residual node embedding $\Delta h_i^{l+1}$, is calculated by a Hadamard product between the runtime angle information and the aggregated scalar messages with a gated residual connection:

$$\Delta h_i^{l+1} = \left\langle W_t^l \vec{v}_i^{\,l}, W_s^l \vec{v}_i^{\,l} \right\rangle \odot W_{Angle}^l m_i^l + W_{res}^l m_i^l \qquad (28)$$

To compute the residual edge embedding $\Delta f_{ij}^{l+1}$, we perform the Hadamard product of the runtime dihedral information with the transformed edge embedding:

$$\Delta f_{ij}^{l+1} = \left\langle \mathrm{Rej}_{\vec{r}_{ij}} \left( W_{Rt}^l \vec{v}_i^{\,l} \right), \mathrm{Rej}_{\vec{r}_{ji}} \left( W_{Rs}^l \vec{v}_j^{\,l} \right) \right\rangle \odot \mathrm{Dense}_{Dihedral}^l \left( f_{ij}^l \right) \qquad (29)$$

After the residual hidden representations are calculated, we add them to the original input of block $l$ and feed them to the next block.

A comprehensive version that includes improper angles is depicted in Supplementary Methods.

**The output block.** Following PaiNN[18], we update the scalar embedding and vector embedding of nodes with multiple gated equivariant blocks:

$$t_i^l = \mathrm{Dense}_{o_2}^l \left( \left[ \left\| W_{o_1}^l \vec{v}_i^{\,l} \right\|, h_i^l \right] \right) \qquad (30)$$

$$h_i^{l+1} = W_{o_3}^l t_i^l \qquad (31)$$

$$\vec{v}_i^{\,l+1} = W_{o_4}^l \vec{v}_i^{\,l} \odot W_{o_5}^l t_i^l \qquad (32)$$

where $[\cdot, \cdot]$ is the tensor concatenation operation. The final scalar embedding $h_i^l \in \mathbb{R}^{N \times 1}$ and vector embedding $\vec{v}_i^{\,l} \in \mathbb{R}^{N \times 3 \times 1}$ are used to predict various molecular properties.

On QM9, the molecular dipole is calculated as follows:

$$\mu = \left| \sum_{i=1}^N \vec{v}_i^{\,l} + h_i^l \left( \vec{r}_i - \vec{r}_c \right) \right| \qquad (33)$$

where $\vec{r}_c$ denotes the center of mass. Similarly, for the prediction of electronic spatial extent $\langle R^2 \rangle$, we use the following equation:

$$\langle R^2 \rangle = \sum_{i=1}^N h_i^l \left| \vec{r}_i - \vec{r}_c \right|^2 \qquad (34)$$

For the remaining 10 properties $y$, we simply aggregate the final scalar embedding of nodes as follows:

$$y = \sum_{i=1}^N h_i^l \qquad (35)$$

For models trained on the molecular dynamics datasets including MD17, revised MD17, and Chignolin, the total potential energy is obtained as the sum of the final scalar embedding of the nodes. As an energy-conserving potential, the forces are then calculated using the negative gradients of the predicted total potential energy with respect to the atomic coordinates:

$$E = \sum_{i=1}^N h_i^L \qquad (36)$$

$$\vec{F}_i = - \nabla_i E \qquad (37)$$

## Statistics and reproducibility

For the QM9 dataset, we randomly split it into 110,000 samples as the train set, 10,000 samples as the validation set, and the rest as the test set by following the previous studies[18,19]. For the Molecule3D and OGB-LSC PCQM4Mv2 datasets, the splitting has been provided in their paper[32,33].

To evaluate the effectiveness of ViSNet in simulation data, ViSNet was trained on MD17 and rMD17 with a limited data setting, which consists of only 950 uniformly sampled conformations for model training and 50 conformations for validation for each molecule. For the MD22 dataset, we use the same number of molecules as in ref. [30] for training and validation, and the rest as the test set.

Furthermore, the whole Chignolin dataset was randomly split into 80%, 10%, and 10% as the training, validation, and test datasets. Six representative conformations were picked from the test set for illustration.

## Experimental settings

For the QM9 dataset, we adopted a batch size of 32 and a learning rate of 1e−4 for all the properties. For the Molecule3D dataset, we adopted a larger batch size of 512 and a learning rate of 2e−4. For the OGB-LSC PCQM4Mv2 dataset, we trained our model in a mixed 2D/3D mode with a batch size of 256 and a learning rate of 2e−4. The mean squared error (MSE) loss was used for model training. For the molecular dynamic dataset including MD17, rMD17, MD22, and Chignolin, we leveraged a combined MSE loss for energy and force prediction. The weight of energy loss was set to 0.05. The weight of force loss was set to 0.95. The batch size was chosen from 2, 4, 8 due to the GPU memory and the learning rate was chosen from 1e−4 to 4e−4 for different molecules. The cutoff was set to 5 for small molecules in QM9, MD17, rMD17, and Molecule3D, and changed to 4 for Chignolin in order to reduce the number of edges in the molecular graphs. For the MD22 dataset, the cutoff of relatively small molecules was set to 5, that of bigger molecules was set to 4. Cutoff was not used in the OGB-LSC PCQM4Mv2 dataset. We used the learning rate decay if the validation loss stopped decreasing. The patience was set to 5 epochs for Molecule3D, 15 epochs for QM9, and 30 epochs for MD17, rMD17, MD22, and Chignolin. The learning rate decay factor was set to 0.8 for these models. Training is stopped if a maximum number of epochs is reached, or the validation loss does not improve for a maximum number of early stopping patience. The ViSNet model trained on the molecular dynamic datasets and Molecule3D had 9 hidden layers and the embedding dimension was set to 256. We used a larger model for the QM9 dataset, i.e., the embedding dimension changed to 512. For the OGB-LSC PCQM4Mv2 dataset, we use the 12-layer and 768-dimension Transformer-M[37] as the backbone. More details about the hyperparameters of ViSNet can be found in Supplementary Table S4. Experiments were conducted on NVIDIA 32G-V100 GPUs.

## Reporting summary

Further information on research design is available in the Nature Portfolio Reporting Summary linked to this article.

## Data availability

All relevant data supporting the key findings of this study are available within the article and its Supplementary Information files. MD17 dataset [http://www.quantum-machine.org/gdml/data/npz], MD22 dataset [http://www.quantum-machine.org/gdml/data/npz], rMD17 dataset [https://archive.materialscloud.org/record/file?filename=rmd17.tar.bz2&record_id=466], QM9 dataset [https://deepchemdata.s3-us-west-1.amazonaws.com/datasets/molnet_publish/qm9.zip], Molecule3D dataset [https://github.com/divelab/MoleculeX/tree/molx/Molecule3D], OGB-LSC PCQM4Mv2 dataset [https://ogb.stanford.edu/docs/lsc/pcqm4mv2] and Chignolin dataset [https://github.com/microsoft/AI2BMD/tree/ViSNet/chignolin_data]. Source data are provided with this paper.

## Code availability

Most experiments were run with Python with version 3.9.15, Pytorch with version 1.11.0, Pytorch Geometric with version 2.1.0, and Pytorch Lightning with version 1.8.0. The code used to reproduce our results is available at https://github.com/microsoft/AI2BMD/tree/ViSNet[57]. Matplotlib and Seaborn were used for plotting figures.

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

## Acknowledgements

We would like to express our sincere gratitude to S. Chmiela, H.E. Sauceda, K.R. Müller, and A. Tkatchenko, for their invaluable assistance in performing the simulations and analyzing the vibrational spectra. Their extensive expertise and knowledge greatly contributed to the completion of the supplementary experiments, making our manuscript more solid.

## Author contributions

T.W. led, conceived, and designed the study. T.W. is the lead contact. Y.W., S.L., X.H., and M.L. conducted the work when they were visiting Microsoft Research. S.L., Y.W., and T.W. carried out algorithm design. Y.W., S.L., X.H., and T.W. carried out experiments, evaluations, analysis, and visualization. Y.W. and S.L. wrote the original manuscript. T.W., X.H., M.L., Z.W., and B.S. revised the manuscript. N.Z. and T.-Y.L. contributed to the writing. All authors reviewed the final manuscript.

## Competing interests

T.W., B.S., and T.-Y.L. have been filing a patent on ViSNet model. The remaining authors declare no competing interests.
