## [Peer Review File · Nature Communications]

Enhancing geometric representations for molecules with equivariant vector-scalar interactive message passingReviewer #1 (Remarks to the Author):

The authors present a new deep learning potential called ViSNet, which leverages the proposed Runtime Geometry Calculation module and an effective vector-scalar interactive message-passing mechanism. The resulting model sometimes improves the performance on several benchmarks of molecular property and force/energy predictions, although this improvement is rather small and not systematic. For example, the overall accuracy of the reference data in the (rev)MD17 dataset is not higher than 0.3 kcal/mol/molecule. Getting errors below this threshold is unphysical and unnecessary. For the MD22 dataset (for some reason hidden in the supplement), the ViSNet improvement does look noticeable. However, no analysis is done on the dynamics of MD22 molecules. Here also one has to keep in mind that the accuracy of the reference data is probably not better than 1-2 kcal/mol/molecule.

The different ingredients of the proposed model (equivariance, spherical harmonics expansions, physically-motivated representations based on angles and torsions, energy conservation) have been addressed by other models that the authors cite in their work. While ViSNet might be a nice combination of the different ingredients, it does not strike me as a substantial advance worthy of publication in Nature Communications. In addition, the manuscript will be mainly of interest to experts in machine learning methods, since the focus is largely on error metrics, rather than on novel insights that could spark the interest of the molecular modeling community.

The application of the proposed method to larger molecular systems lacks the necessary rigor and thoroughness in its current form. Additionally, the writing lacks clarity in several sections, with some facts stated without proper discussion or analysis. This is a more detailed list of issues to be addressed:

1. The obtained results from MD17/revMD17 simulations do not yield any significant surprises or insights. However, the presentation in Section 2.3 creates the impression that these simulations are being conducted for the first time within the scope of this paper. Moreover, the comparison with only DFT (in the main text and in Table S3, Fig. S3) implies that it is the sole viable option, disregarding the existence of other machine learning force fields (MLFFs) that can offer similar accuracy and computational time savings. To provide a more comprehensive perspective, it would be appropriate to acknowledge the availability of other MLFFs that can deliver comparable results in terms of both accuracy and time efficiency. In fact, I suspect that sGDML kernel approach is faster both for training and evaluation in these datasets, see for example <https://pubs.rsc.org/en/content/articlehtml/2021/sc/d1sc03564a>. Also, comparing distance distributions is not a stringent test of the ML potential. One should do a stringent analysis of free energy surfaces, vibrational spectra, and occupation probabilities of different minima (see <https://www.nature.com/articles/s41467-018-06169-2>).

2. The authors performed a simulation of the Chignolin peptide for 5 ps. This is not enough to demonstrate "usefulness for real-world applications". It has been shown that severe pathologies might become apparent only after hundreds of ps or even several ns (10.1088/2632-2153/ac9955, 10.48550/arXiv.2209.03985, 10.48550/arXiv.2210.07237). To address this, it would be beneficial to perform longer MD simulation of Chignolin and analyze the resulting dynamics.

The interpretation of the root mean square fluctuations of each atom averaged over 10 runs (Fig. 5d) is not clear. It would be more informative to provide root-mean square deviation for the whole structure as a function of time for a single extended simulation. Reliable dynamics should explore the potential energy surface, including structures similar to the 10 selected ones. Furthermore, the comparison with DFT (Fig. 5e) as a force norm for the entire molecule is not informative. It would be more useful to demonstrate this comparison using violin plots or by providing MAE/RMSE values of each component of the atomic forces.

The comparison made only with an empirical force field simulation of Chignolin holds limited value and does not showcase the true strength of the proposed method. Many existing architectures have the potential to demonstrate better performance on this system. Without this comparison, the claim that the proposed model "alleviates the dilemma between computational costs and sufficient utilization of geometric information" can only be applied to small molecules.

Also, there seems to be a contrast in performance between the new peptide and the MD22 results. The reported energy mean absolute error of 3.62 kcal/mol for a 166-atom system and the very small errors for MD22 (e.g., ~0.2 kcal/mol for a 118-atom system and ~0.6 kcal/mol for a 370-atom system) raise questions.

3. "To alleviate this problem, inspired by [16], we propose runtime geometry calculation (RGC), which uses an equivariant vector representation (termed as "direction unit") for each node to preserve its geometric information."

"Furthermore, according to the many-body expansion theory [42–44], the potential energy of the whole system equals to the potential of each single atom plus the energy corrections from two-bodies to many-bodies. ... ViSNet encodes the triplet and quadruplet interactions in a single block, which empowers the model to have much more powerful representation ability"

The inclusion of a discussion on the completeness of the proposed representation would enhance the paper's clarity and depth. It is important to note that neither many-body representations, such as those using angles or dihedral angles, nor equivariant representations based on multipole expansions are considered complete. Previous studies (10.1103/PhysRevLett.125.166001) have shown that even with the inclusion of up to 4-body terms, there exist structures containing as few as eight atoms that cannot be distinguished. Also, the clustering in Fig. 6 requires further discussion, C11 and C12 atoms are indeed of different natures but are depicted in one cluster, which shows that the description might not be complete.

4. "Based on a Chignolin dataset consisting of about 10,000 conformations that sampled by replica exchange MD and calculated at DFT level by Gaussian 16 in our another study."

"We trained ViSNet and compared it with molecular mechanics (MM)"

Currently, there is a lack of information on the specific methodologies and parameters used for both DFT and MM simulations. It is important to include these details to ensure transparency and reproducibility. The authors should ensure that they cite the computational package used for their calculations, such as Gaussian 16, to acknowledge the software utilized in their research.

Other minor comments:

5. "To alleviate this problem, inspired by [16], we propose runtime geometry calculation (RGC), which uses an equivariant vector representation (termed as "direction unit") for each node to preserve its geometric information."

It is unclear from the text what was taken from Ref. 16 and what is different.

6. "thus significantly accelerating model training and inference while reducing the memory consumption."

To support the statement authors should include specific details about the training time and memory consumption for a challenging system or systems. While the authors report time latency, it is unclear for which molecule this measurement was taken. If it corresponds to 3BPA as in Ref. 22, it would be beneficial to provide error metrics for ViSNet as well as other models specifically for this molecule. Would this time latency trend be the same for increasingly larger systems?

7. "ViSNet also has won PCQM4Mv2 track in the OGB-LCS@NeurIPS2022 competition (<https://ogb.stanford.edu/neurips2022/results/>)".

ViSNet is listed second on the scoreboard, MAE in Figure S1 (0.0771) and on the website (0.0723) is different.

8. "the distance between mainchain O on Y2 and mainchain N on G6", "mainchain O on Y2 and mainchain N on G6) and dE4–T7"

It is unclear what are the atoms to which authors refer in the text.

9. "Prior works have shown the effectiveness of incorporating geometric features, such as angles"
This sentence requires citation.

10. In Table 1 and in some places in the text there are improvements measured in percentages with 3-4 significant digits. Is this justified?

11. "Table 2 footnote. ViSNet can achieve better results with longer convergence time."
Isn't this true for all models? Also, authors should state what is the convergence criteria in their case in the Methods section.

12. "The consistent potential energy surfaces suggest that ViSNet can well recover the kinetic properties and the conformational space from the simulation trajectories, indicating the usefulness of ViSNet for real molecular dynamics simulation."
What kinetic properties do authors refer to?

13. In Fig. 5 all six examples show errors smaller than E_MAE of 3.62 kcal/mol. It would be more instructive to show challenging examples with high errors as well. Also, reporting MAE for one structure seems strange.

14. It would also be beneficial to provide additional clarification in the text, particularly in Section 2.1, to explicitly explain the aspects of ViSNet's approach that contribute to its improved performance.

15. "It is worth noting that ViSNet is an energy-conserving potential, i.e., the predicted atomic forces are derived from the negative gradients of the potential energy with respect to the coordinates [23]."
[23] Chmiela, S., Sauceda, H. E., Muller, K.-R. & Tkatchenko, A. Towards exact molecular dynamics simulations with machine-learned force fields. Nature communications 9, 1–10 (2018).

Citation is missing to Ref. 28.

16. "In addition, considering that angle and dihedral are important potential terms in empirical force fields, the interpretability of the operations in the RGC strategy provides some insights in constructing hybrid force fields by combining empirical terms with deep learning."

I do not understand how the presented interpretability can be used, if at all.

17. "triplet and quadruplet interactions" are confusing terms.

Reviewer #2 (Remarks to the Author):

In the past decade, with the rapid development of deep learning, revolutionary breakthroughs were achieved in the fields of machine learning and artificial intelligence. More recently, it has permeated into various fields of science, among which an important topic is the machine-learning-based force field. In the submitted manuscript, the authors reported an exciting progress in this direction. More specifically, they we propose a runtime geometry calculation (RGC) to effectively encode the geometric (conformational) information of molecules and Vector-Scalar interactive graph neural Network (ViSNet) as a machine learning model for molecular force field. The ViSNet shows remarkable performances in a few tested cases. For example, it achieved the top winners of PCQM4Mv2 track in the OGB-LCS@NeurIPS2022 competition, and it outperforms the state-of-the-art approaches on the molecules in the MD17 dataset, with only 0.7% samples (i.e., 950 samples for model training). This is an exciting achievement since deep learning usually requires a lot of data. It may have a profound impact in the field of machine-learning-based force field. I recommend the manuscript to be published after proper revision:

- (1) According to the usual practice, Microsoft Research should be the first affiliation.
- (2) p6: "MD17 consists of the MD trajectories of 7 small organic molecules": 17 but not 7 small organic molecules, or 7 small organic molecules and 10 other molecules?
- (3) Eq.(3): the dot "." should be deleted to avoid possible misunderstanding that it is an inner product.
- (4) Chignolin is an (artificial) mini-protein. It is not appropriate to call it protein.

Reviewer #3 (Remarks to the Author):

Summary:

The paper introduces a graph neural network that incorporates a new geometric feature extraction module that models multi-body interaction in atom graphs with reduced computational costs. The proposed model achieves state-of-the-art performance on several molecular dynamics and property prediction tasks.

Geometric information such as angles, dihedral torsions, and improper angles are essential for learning a good molecular representation of downstream tasks like fitting potential energy surfaces. However, they are usually expensive to model. This paper proposes a framework called Runtime Geometry Calculation that is capable of extracting the above geometric information with better efficiency compared to several existing models. The authors also validate the efficacy of the learned embedding by doing a clustering. While the proposed model is a worthy addition to the existing GNN architectures for molecular machine learning, there are several points in the current presentation that can be further improved and clarified.

Comment:

The Scalar2Vec and Vec2Scalar modules introduced in the Method section of this paper are in fact closely related to PaiNN and TorchMD-NET. The current description makes the contribution of this work entangled with these existing works, for example, attention is used in both this work and TorchMD-NET. The authors can elaborate more on the difference between their models and existing architectures, and potentially provide more motivations for the architectural design. Under Table 2, the authors state that "ViSNet can achieve better results with longer convergence time." Does it mean ViSNet converges slower than the other models? If so, does this imply ViSNet generally requires more training iterations than other models, especially compared with TorchMD-NET, which has a similar message passing protocol but without the RGC module. Following up the last question, how is the computational cost of ViSNet compared to PaiNN/TorchMD-NET?

Response to Reviewer #1

The authors present a new deep learning potential called ViSNet, which leverages the proposed Runtime Geometry Calculation module and an effective vector-scalar interactive message-passing mechanism. The resulting model sometimes improves the performance on several benchmarks of molecular property and force/energy predictions, although this improvement is rather small and not systematic. For example, the overall accuracy of the reference data in the (rev)MD17 dataset is not higher than 0.3 kcal/mol/molecule. Getting errors below this threshold is unphysical and unnecessary. For the MD22 dataset (for some reason hidden in the supplement), the ViSNet improvement does look noticeable. However, no analysis is done on the dynamics of MD22 molecules. Here also one has to keep in mind that the accuracy of the reference data is probably not better than 1-2 kcal/mol/molecule.

Response: Thank you for taking the time to carefully review our manuscript. We highly value your expertise, and your critical feedback is instrumental in enhancing the quality of our work. Considering your comments, we have systematically complemented experiments, carefully revised the manuscript and made additional adjustments that we believe satisfactorily address the concerns. We trust that these revisions will provide a clearer and more accurate picture of our work. Thanks for your professional input, which has helped us a lot to shape the manuscript into a more comprehensive study for a wider range of readers.

First, with respect to performance enhancement, we clarify that ViSNet has achieved comparable or even more significant performance gains than previous studies achieved before. For example, Allegro (Nat. Comm., 14(1), 579.) reduced the energy mean absolute error (MAE) by 0.2 meV compared to its previous work NequIP (Nat. Comm. , 13(1), 2453) for 1 of 10 molecules in the rMD17 dataset. In contrast, this study has delivered the best energy prediction results for all molecules in the MD17 dataset and for 9 of 10 molecules in the rMD17 dataset, respectively. Furthermore, the performance gains are also significant on MD22 dataset, as these larger molecules tend to possess more intricate 3D structures. Although the benchmark dataset itself may have some accuracy limits, the goal of model evaluation on the widely used benchmark is to approximate the labels as closely as possible and fairly evaluate model performance among state-of-the-art algorithms, facilitating increasingly accurate predictions as the dataset's accuracy improves.

Second, we clarify that putting the MD22 evaluation table in the supplementary material was primarily due to the limitations of the number of figures and tables in the submission guideline. We have moved the table to the revised manuscript for better illustration.

Third, per your suggestion, we have performed MD simulations for Ac-Ala3-NHMe on the MD22 dataset and ethanol in MD17 dataset. The simulations were conducted by ViSNet, sGDML and quantum simulation at density functional theory (DFT) level respectively, and the vibrational spectra and Ramachandran plot were analyzed. As shown in Fig. 1 in the response letter and Section 2.3 in the revised manuscript, the results indicate that both sGDML and ViSNet can explore and exhibit reasonable molecular properties and have good alignments with DFT.

Fig. 1 **Analysis of MD simulations driven by DFT, sGDML and ViSNet.** (a) Vibrational spectra analysis of Ac-Ala3-NHMe in MD22 dataset. (b) Vibrational spectra analysis of Ethanol in MD17 dataset. (c) Analysis of the probability distribution of hydroxyl angles of ethanol. (d) to (f) From left to right, the Ramachandran plots of Ac-Ala3-NHMe predicted or calculated by DFT, sGDML and ViSNet.

Additionally, ViSNet is not solely tailored for molecular dynamics applications. The enhancements that ViSNet offers in molecular property prediction datasets are also significant. For instance, within the QM9 dataset, ViSNet achieves the best performance in 9 out of 12 tasks, while prior works typically attain top results in merely two or three tasks. The winner in the OGB-LSC competition further demonstrates ViSNet's capacity to generalize effectively to previously unseen test sets.

The different ingredients of the proposed model (equivariance, spherical harmonics expansions, physically-motivated representations based on angles and torsions, energy conservation) have been addressed by other models that the authors cite in their work. While ViSNet might be a nice combination of the different ingredients, it does not strike me as a substantial advance worthy of publication in Nature Communications. In addition, the manuscript will be mainly of interest to experts in machine learning methods, since the focus is largely on error metrics, rather than on novel insights that could spark the interest of the molecular modeling community.

Response: We clarify that the aspects of equivariance and energy conservation discussed in the manuscript provide background knowledge to facilitate the reader's understanding while the novelty and distinctiveness of ViSNet can be summarized through the following aspects:

1. We developed a Runtime Geometry (both for dihedral and improper angle) Calculation (RGC) module with linear time complexity. Building upon the vectorized representation of angles proposed in PaiNN and ET, we further extended the vectorized representation and operations

- to 4-body interactions. This approach enables us to represent all bonded terms in classical MD as hidden representations and model operations, consequently enhancing model interpretability.
2. We designed a novel message-passing mechanism called ViS-MP for edges and mapped 4-body interactions to edges. The proposed ViS-MP encodes and models geometric information for both nodes and edges, contributing significantly to performance gains. To the best of our knowledge, we are the first to implement four-body interactions, particularly dihedral torsion angles, with linear complexity, compared to GemNet. Other work in the ACE series can only describe 4-body interactions within one cluster, and their work does not encompass dihedral angles.
 3. Additionally, we incorporated the inner product of high-order representations ($L=2$) to improve model performance without cumbersome Clebsch-Gordan (CG) product operations. Our research found that the inner product of high-order representations can enhance model performance without substantial computational overheads, whereas other methods employ complex CG products. This finding can largely reduce the computational complexity of the model which could be applied to larger systems.

We believe that the introduction of dihedral angles in ViSNet is highly intriguing. We are confident that improvements in performance and computational efficiency and the application in MD simulations will be of interest to both the scientific community and the computer science community.

The application of the proposed method to larger molecular systems lacks the necessary rigor and thoroughness in its current form. Additionally, the writing lacks clarity in several sections, with some facts stated without proper discussion or analysis. This is a more detailed list of issues to be addressed:

1. The obtained results from MD17/revMD17 simulations do not yield any significant surprises or insights. However, the presentation in Section 2.3 creates the impression that these simulations are being conducted for the first time within the scope of this paper. Moreover, the comparison with only DFT (in the main text and in Table S3, Fig. S3) implies that it is the sole viable option, disregarding the existence of other machine learning force fields (MLFFs) that can offer similar accuracy and computational time savings. To provide a more comprehensive perspective, it would be appropriate to acknowledge the availability of other MLFFs that can deliver comparable results in terms of both accuracy and time efficiency. In fact, I suspect that sGDML kernel approach is faster both for training and evaluation in these datasets, see for example <https://pubs.rsc.org/en/content/articlehtml/2021/sc/d1sc03564a>. Also, comparing distance distributions is not a stringent test of the ML potential. One should do a stringent analysis of free energy surfaces, vibrational spectra, and occupation probabilities of different minima (see <https://www.nature.com/articles/s41467-018-06169-2>).

Response: We clarify that we did not intend to imply that we were the first to conduct the simulation. In fact, several methods have been employed to perform simulations, with sGDML being the most notable pioneer among them. The purpose of our simulation experiments is to further evaluate the utility of our model in real applications. We have introduced and cited a series of works on simulation in the paper accordingly as follows:

“Introduction: Gradient-domain machine learning (GDML) [1] constructs accurate molecular force fields using conservation of energy and limited samples from ab initio molecular dynamics trajectories, enabling cost-effective simulations while maintaining accuracy. Symmetric GDML (sGDML) [2] further improves force field construction by incorporating physical symmetries, achieving CCSD(T)-level accuracy for flexible molecules. An exact iterative approach (Global sGDML) [3] extends sGDML to global force fields for molecules with several hundred atoms, maintaining correlations of atomic degree and accurately describing complex molecules and materials.”

“Section 2.3: Moreover, compared to DFT, numerous groundbreaking machine learning force fields (MLFFs), including sGDML [2], ANI [4], DPMD [5], and PhysNet [6] have proven their exceptional speeds in MD simulations. Similar to such algorithms, ViSNet also exhibited significant computational cost reduction compared to DFT as shown in Supplementary Fig. S3 and Table S2.”

[1] Chmiela, Stefan, et al. "Machine learning of accurate energy-conserving molecular force fields." *Science advances* 3.5 (2017): e1603015.

[2] Chmiela, Stefan, et al. "Towards exact molecular dynamics simulations with machine-learned force fields." *Nature communications* 9.1 (2018): 3887.

[3] Chmiela S, Vassilev-Galindo V, Unke O T, et al. Accurate global machine learning force fields for molecules with hundreds of atoms[J]. *Science Advances*, 2023, 9(2): eadf0873.

[4] Smith, Justin S., Olexandr Isayev, and Adrian E. Roitberg. "ANI-1: an extensible neural network potential with DFT accuracy at force field computational cost." *Chemical science* 8.4 (2017): 3192-3203.

[5] Zhang, Linfeng, et al. "Deep potential molecular dynamics: a scalable model with the accuracy of quantum mechanics." *Physical review letters* 120.14 (2018): 143001.

[6] Unke, Oliver T., and Markus Meuwly. "PhysNet: A neural network for predicting energies, forces, dipole moments, and partial charges." *Journal of chemical theory and computation* 15.6 (2019): 3678-3693.

Following your suggestions, we refer to the sGDML paper for a more in-depth analysis of the MD simulations for the ethanol molecule in the MD17 dataset and Ac-Ala3-NHMe in the MD22 dataset. For ethanol, we examined its vibrational spectra and the probability distribution of dihedral angles. For Ac-Ala3-NHMe, we evaluated the vibrational spectra and Ramachandran plot. For a fair and rigor comparison, we performed the simulations using force field driven by ViSNet, sGDML, and DFT in the exact same environment (ASE) software, respectively. Similar results demonstrated that both our approach and sGDML can perform stable and efficient simulations. The slight differences with DFT may arise from the fact that the specific settings in our DFT calculation and those in the previous dataset generation are slightly different (Fig. 1 in the response letter). We are very grateful to the authors of sGDML for their assistance in performing these experiments. We have added an Acknowledgement Section as follows:

“We would like to express our sincere gratitude to Chmiela, S., Sauceda, H.E., Müller, K.R. and Tkatchenko, A, for their invaluable assistance in performing the simulations and analyzing the vibrational spectra. Their extensive expertise and knowledge greatly contributed to the completion of the supplementary experiments, making our manuscript more solid.”

Once again, the additional experiments you suggested have further improved the reliability of our model. (See Section 2.3 in the revised manuscript and Fig. 1 in the response letter for more details.)

2. The authors performed a simulation of the Chignolin peptide for 5 ps. This is not enough to demonstrate “usefulness for real-world applications”. It has been shown that severe pathologies might become apparent only after hundreds of ps or even several ns (10.1088/2632-2153/ac9955, 10.48550/arXiv.2209.03985, 10.48550/arXiv.2210.07237). To address this, it would be beneficial to perform longer MD simulation of Chignolin and analyze the resulting dynamics.

The interpretation of the root mean square fluctuations of each atom averaged over 10 runs (Fig. 5d) is not clear. It would be more informative to provide root-mean square deviation for the whole structure as a function of time for a single extended simulation. Reliable dynamics should explore the potential energy surface, including structures similar to the 10 selected ones. Furthermore, the comparison with DFT (Fig. 5e) as a force norm for the entire molecule is not informative. It would be more useful to demonstrate this comparison using violin plots or by providing MAE/RMSE values of each component of the atomic forces.

The comparison made only with an empirical force field simulation of Chignolin holds limited value and does not showcase the true strength of the proposed method. Many existing architectures have the potential to demonstrate better performance on this system. Without this comparison, the claim that the proposed model "alleviates the dilemma between computational costs and sufficient utilization of geometric information" can only be applied to small molecules.

Also, there seems to be a contrast in performance between the new peptide and the MD22 results. The reported energy mean absolute error of 3.62 kcal/mol for a 166-atom system and the very small errors for MD22 (e.g., ~0.2 kcal/mol for a 118-atom system and ~0.6 kcal/mol for a 370-atom system) raise questions.

Response: We are grateful for your valuable feedback. In accordance with your suggestions, we have restructured the experiment and extended the simulation time for Chignolin by 20 times, resulting in a 100ps simulation. Following your suggestion, we replaced the original analysis for RMSF with an analysis of RMSD for 10 trajectories. Furthermore, in response to your comments, we have removed the force norm comparison and displayed the MAE values of each component of the atomic forces over the simulation. It is evident that our ViSNet exhibits relatively low MAE across all force directions (Fig. 2 in the response letter).

Furthermore, per your suggestion, we added state-of-the-art methods including NequIP, Allegro, PaiNN, ET and GemNet-OC for performance comparison on Chignolin. On this challenging dataset,

ViSNet has a better performance on energy and force metrics (both MAE and R^2) than the other methods, illustrating the superiority of our algorithm. (See Fig. 2, Fig. 3 and Fig. 4 in the response letter and Section 2.4 in the revised manuscript for a more detailed description.)

In response to your final concern, we clarify that Chignolin dataset is challenging, as the conformations were extracted from multiple trajectories of replica exchange molecular dynamics (REMD) simulations. By merging MD simulation with the Monte Carlo algorithm, the REMD method can effectively surmount high energy barriers and sufficiently sample the conformational space [1]. As a result, various conformations within the conformational space are present in both the training set and the test set, which may lead to a larger MAE of the model compared to those in MD22 dataset.

[1]: Qi et al. Replica Exchange Molecular Dynamics: A Practical Application Protocol with Solutions to Common Problems and a Peptide Aggregation and Self-Assembly Example.

Fig. 2 Applications of ViSNet for Chignolin conformational space evaluation and MD simulations. (a) The energy landscape of Chignolin sampled by REMD. The x-axis of the landscape is the distance between carbonyl oxygen on D3 backbone and nitrogen on G7 backbone, while the y-axis is the distance between carbonyl oxygen on E5 backbone and nitrogen on T8 backbone. 6

structures were then selected for visualization. Each structure is shown as cartoon and residues are depicted in sticks. The histograms show the absolute error between the energy difference predicted or calculated by MLFFs or MM, and the ground truth calculated by DFT on the corresponding structure. (b) The average root mean square deviation (RMSD) of the Chignolin trajectories simulated by ViSNet calculated from 10 different trajectories. The shaded areas indicate the standard deviation range. (c) The MAE of each component of atomic forces during the simulations driven by ViSNet. The ground truth energies and forces were calculated using Gaussian 16. The shaded areas indicate the standard deviation range.

Fig. 3 The energy correlations between the ground truth calculated by DFT and predictions or calculations by MLFFs and molecular mechanics (MM) respectively on the test dataset. The results are predicted or calculated by ViSNet, ET, PaiNN, GemNet-OC, NequIP, Allegro and MM, respectively. The corresponding distributions of energy predictions or calculations as well as the ground truth are also shown in each panel.

Fig. 4 The forces correlations between the ground truth calculated by DFT and predictions or calculations by MLFFs and molecular mechanics (MM) respectively on the test dataset. From top to bottom, the results are predicted or calculated by ViSNet, ET, PaiNN, GemNet-OC, NequIP, Allegro and MM, respectively. From left to right, the results are the component of forces in the three

directions x, y, z , respectively. The corresponding distributions of forces predictions or calculations as well as the ground truth are also shown in each panel.

3. “To alleviate this problem, inspired by [16], we propose runtime geometry calculation (RGC), which uses an equivariant vector representation (termed as “direction unit”) for each node to preserve its geometric information.”

“Furthermore, according to the many-body expansion theory [42–44], the potential energy of the whole system equals to the potential of each single atom plus the energy corrections from two-bodies to many-bodies. ... ViSNet encodes the triplet and quadruplet interactions in a single block, which empowers the model to have much more powerful representation ability”

The inclusion of a discussion on the completeness of the proposed representation would enhance the paper’s clarity and depth. It is important to note that neither many-body representations, such as those using angles or dihedral angles, nor equivariant representations based on multipole expansions are considered complete. Previous studies (10.1103/PhysRevLett.125.166001) have shown that even with the inclusion of up to 4-body terms, there exist structures containing as few as eight atoms that cannot be distinguished. Also, the clustering in Fig. 6 requires further discussion, C11 and C12 atoms are indeed of different natures but are depicted in one cluster, which shows that the description might not be complete.

Response: Thank you for your valuable comments. We appreciate your suggestion to incorporate a discussion on completeness, which will undoubtedly improve the paper’s clarity and depth. We clarify that ViSNet may not be complete as a corner case was shown in Fig. 8 in the revised manuscript. In light of your feedback, we have included a discussion in the manuscript about the completeness as follows:

“In addition, ViSNet’s universality or completeness is not validated by the Geometric Weisfeiler-Leman (GWL) test [1] due to the inner product operation, which is computationally efficient but fails to distinguish certain atom reflection structures with the same angular information. To pass counterexamples or the GWL test, incorporating the CG-product with higher-order spherical harmonics is necessary in the future study.”

[1] Joshi, Chaitanya K., et al. “On the expressive power of geometric graph neural networks.” *arXiv preprint arXiv:2301.09308* (2023).

Additionally, we plan to develop models capable of passing the 3D-Weisfeiler-Lehman test as our future research endeavors. Thank you.

4. “Based on a Chignolin dataset consisting of about 10,000 conformations that sampled by replica exchange MD and calculated at DFT level by Gaussian 16 in our another study.”

“We trained ViSNet and compared it with molecular mechanics (MM)”

Currently, there is a lack of information on the specific methodologies and parameters used for both DFT and MM simulations. It is important to include these details to ensure transparency and reproducibility. The authors should ensure that they cite the computational package used for their calculations, such as Gaussian 16, to acknowledge the software utilized in their research.

Response: The details of the Chignolin dataset were introduced in our another study (*J. Chem. Phys.*, 159 (3): 035101). Per your suggestion, to maintain consistency and ensure all necessary information is available, we have added the detailed settings for both DFT and MM simulations in the Supplementary Materials as follows:

“The replica exchange MD (REMD) simulations [1] utilized an initial structure sourced from the Protein Data Bank (PDB ID: 5AWL) [2]. Water molecules present in this crystal structure were removed. For the atomic interactions of Chignolin, we applied the FF19SB force field [3] within a generalized Born implicit solvent model. This model incorporated the second modification of the Bondi Van der Waals radii set [4]. To preserve chiral integrity at elevated temperatures during REMD simulations, we used the makeCHIR_RST tool in Amber 20 [5] to generate a chiral restraint file.

The system underwent an energy minimization, consisting of 500 steepest descent cycles followed by 500 conjugate gradient cycles. This was succeeded by 200ps of equilibration runs across temperatures from 300K to 1000K, initialized with randomized velocities. The resultant structure from this equilibration was the base for REMD simulations at respective temperatures. In production, every replica ran for 2ps before exchanging with its adjacent temperature. This resulted in a total of 5,000 exchanges per production run, culminating in an 80ns simulation over 8 replica temperatures. Sampling of the trajectory was done at 0.4ps intervals. The simulations were accomplished via Amber20 sander MPI version.

From the REMD trajectory, 10,000 uniformly distributed points were chosen to construct the input for Gaussian 16 [6]. Each conformation’s potential energy and atomic forces were determined using the M06-2X functional alongside the 6-31G* basis set, supported by a superfine precision integration grid. In summary, the Chignolin dataset incorporated 9,543 converged self-consistent field conformations, complete with total potential energy and atomic forces.

Calculations in molecular mechanics (MM) utilized the FF19SB force field. By extracting target structures and using the tleap program, we generated the necessary topology and coordinate files. The sander program determined the energy of each structure and the forces on individual atoms. Relative energies were derived by comparing each structure’s energy to the one with the minimum energy.”

[1] Qi, R., Wei, G., Ma, B. & Nussinov, R. Replica exchange molecular dynamics: A practical application protocol with solutions to common problems and a peptide aggregation and self-assembly example. In *Peptide self-assembly*, 101–119 (Springer, 2018).

[2] Honda, S. et al. Crystal structure of a ten-amino acid protein. *Journal of the American Chemical Society* 130, 15327–15331 (2008).

[3] Tian, C. et al. ff19sb: Amino-acid-specific protein backbone parameters trained against quantum mechanics energy surfaces in solution. *Journal of chemical theory and computation* 16, 528–552 (2019).

[4] Onufriev, A., Bashford, D. & Case, D. A. Exploring protein native states and large-scale conformational changes with a modified generalized born model. *Proteins: Structure, Function, and Bioinformatics* 55, 383–394 (2004).

[5] Case, D. A. et al. Amber 2021 (University of California, San Francisco, 2021).

[6] Frisch, M. J. et al. Gaussian~16 Revision C.01 (2016). Gaussian Inc. Wallingford CT.

Additionally, we have included citations for the Gaussian 16 and Amber software packages used in our research in the manuscript. Thank you.

Other minor comments:

5. “To alleviate this problem, inspired by [16], we propose runtime geometry calculation (RGC), which uses an equivariant vector representation (termed as “direction unit”) for each node to preserve its geometric information.”

It is unclear from the text what was taken from Ref. 16 and what is different.

Response: Thanks for your comments. As stated earlier, building upon the vectorized representation of angles proposed in PaiNN, we have further extended the vectorized representation and operations to encompass 4-body interactions, such as dihedral torsion and improper. This idea allows us to unify all bonded terms in classic MD with hidden representations and model operations, thereby enhancing the model’s interpretability. We have revised the manuscript to clarify the distinctions between ViSNet and Ref. 16 as follows:

“Notably, beyond employing angular information that has been used in PaiNN [1] and ET [2], ViSNet further considers the dihedral torsion and improper angle calculation with higher geometric tensors.”

“It is important to note that previous studies [1, 2] primarily focused on updating node features, whereas our approach updates both node and edge features during message passing, leading to a more comprehensive geometric representation.”

[1] Schütt, Kristof, Oliver Unke, and Michael Gastegger. "Equivariant message passing for the prediction of tensorial properties and molecular spectra." *International Conference on Machine Learning*. PMLR, 2021.

[2] Thölke, Philipp, and Gianni De Fabritiis. "Equivariant transformers for neural network based molecular potentials." *International Conference on Learning Representations*. 2021.

6. “thus significantly accelerating model training and inference while reducing the memory consumption.”

To support the statement authors should include specific details about the training time and memory consumption for a challenging system or systems. While the authors report time latency, it is unclear for which molecule this measurement was taken. If it corresponds to 3BPA as in Ref. 22, it would be beneficial to provide error metrics for ViSNet as well as other models specifically for this molecule. Would this time latency trend be the same for increasingly larger systems?

Response: Thank you for your comment. To address your concern regarding training time and memory consumption, we have included a comparison of ViSNet with other models, including NequIP, Allegro, GemNet-OC, ET and PaiNN evaluated on the 166-atom Chignolin dataset (see Fig. 5 in the response letter or Fig. 7 in the revised manuscript). We have discussed the computational cost on the challenging Chignolin dataset in Section 2.4 as follows:

“Furthermore, we also made a comprehensive comparison by taking model performance, training time consumption, and the model size all into consideration. ViSNet and other state-of-the-art algorithms such as PaiNN, ET, GemNet-OC, NequIP, and Allegro were analyzed on the Chignolin dataset and shown in Fig. 7. Although ViSNet is marginally slower than ET and PaiNN, it introduces more geometric information, significantly enhancing its performance. Furthermore, when compared to GemNet, which also incorporates dihedral angles, ViSNet's computational cost is significantly more affordable. Similarly, ViSNet proves to be computationally efficient when compared to models employing the CG-product method, such as Allegro and NequIP.”

Regarding the time latency, most of our values were directly obtained from MACE. As stated in MACE, as long as the GPU threads are not fully occupied, time latency remains independent of the number of atoms. Therefore, we did not specifically measure time latency on 3BPA. Instead, we chose to evaluate on the Azobenzene molecule from the MD17 dataset, which is of a similar size to 3BPA.

We hope that these clarifications, along with the additional information and supplementary experiments could address your concerns and further support our claim of accelerated model training and reduced memory consumption.

Fig. 5 The comparison of model performance (Y-axis), training time consumption (X-axis), and model size (volume) among ViSNet (red), PaiNN, ET, GemNet-OC, Allegro, and NequIP (grey) evaluated on the Chignolin dataset.

7. “ViSNet also has won PCQM4Mv2 track in the OGB-LCS@NeurIPS2022 competition (<https://ogb.stanford.edu/neurips2022/results/>)”.

ViSNet is listed second on the scoreboard, MAE in Figure S1 (0.0771) and on the website (0.0723) is different.

Response: We clarify that the differences in the MAE values can be attributed to the fact that they were obtained on different evaluation datasets. In the competition, the test data is not publicly available, so researchers use the validation set for evaluation when comparing methods. Moreover, during the competition, the training and validation sets are combined for training, and model ensemble is employed, resulting in better performance compared to the single model results reported in our manuscript. In addition, the top 3 methods were recognized as the winners of the competition by the organizers. We have modified the sentence as “ViSNet was also one of the winners of PCQM4Mv2 track in the OGB-LCS@NeurIPS2022 competition (<https://ogb.stanford.edu/neurips2022/results/>)”. Thank you.

8. “the distance between mainchain O on Y2 and mainchain N on G6”, “mainchain O on Y2 and mainchain N on G6) and dE4–T7”

It is unclear what are the atoms to which authors refer in the text.

Response: Thank you for highlighting the discrepancy. During the revision, we identified inconsistencies in the definition of our PES parameters. Specifically, the X-axis represents the distance between the carbonyl oxygen on the D3 backbone and the nitrogen on the G7 backbone. Meanwhile, the Y-axis measures the distance between the carbonyl oxygen on the E5 backbone and the nitrogen on the T8 backbone. We've fixed this in our revised manuscript. Thank you.

9. "Prior works have shown the effectiveness of incorporating geometric features, such as angles"
This sentence requires citation.

Response: We appreciate your suggestion. In response to your comment, we have added references to relevant works, such as DimeNet [1] and GemNet [2], which have demonstrated the benefits of gradually incorporating more geometric information to obtain better performance.

[1] Gasteiger, Johannes, Janek Groß, and Stephan Günnemann. "Directional message passing for molecular graphs." *ICLR* 2020.

[2] Gasteiger, Johannes, Florian Becker, and Stephan Günnemann. "Gemnet: Universal directional graph neural networks for molecules." *Advances in Neural Information Processing Systems* 34 (2021): 6790-6802.

10. In Table 1 and in some places in the text there are improvements measured in percentages with 3-4 significant digits. Is this justified?

Response: We appreciate your comment regarding the significant digits. We employ percentages in these instances to represent relative improvements, as the absolute values might be small while the relative improvements are quite significant. Considering that most methods do not use relative improvements for the MD17 dataset, we have removed the comparisons from the tables and retained the relevant descriptions regarding speed in Computational Efficiency part.

11. "Table 2 footnote. ViSNet can achieve better results with longer convergence time."
Isn't this true for all models? Also, authors should state what is the convergence criteria in their case in the Methods section.

Response: The rationale of the statement "ViSNet can achieve better results with longer convergence time" is that, due to limited computational resources, training on the rMD17 dataset was stopped after 3000 epochs, even though there were no signs of overfitting. We think that this may not be suitable for all models, as many could face overfitting issues during training. To avoid any confusion, we removed this footnote.

We recognize the significance of defining convergence criteria in the Methods section and have now incorporated a description of the convergence criteria used in our training as follows:

“Training is stopped if a maximum number of epochs is reached, or the validation loss does not improve for a maximum number of early stopping patience.”

More details about the hyperparameters of ViSNet can be found in Supplementary Table S4.

12. “The consistent potential energy surfaces suggest that ViSNet can well recover the kinetic properties and the conformational space from the simulation trajectories, indicating the usefulness of ViSNet for real molecular dynamics simulation.”

What kinetic properties do authors refer to?

Response: Kinetics properties refer to the change of different states in the PES of small molecules. Based on your previous comments, we have additionally added vibrational spectra about the ethanol and removed kinetic properties for any potential confusion.

13. In Fig. 5 all six examples show errors smaller than E_MAE of 3.62 kcal/mol. It would be more instructive to show challenging examples with high errors as well. Also, reporting MAE for one structure seems strange.

Response: We are grateful for your insightful comment and concur with your perspective. We have showcased typical structures with both low and high errors in Fig. 2 in the response letter. We have discussed these selected structures as follows:

“We randomly selected six structures from different regions of the potential energy surface for visualization. Among them, four structures were predicted by the model with smaller errors than the MAE while the other two with larger errors. Interestingly, all models consistently performed poorly on the structures with high potential energies (low probability of sampling) and performed well on the other structures. This implies that the sampling of conformations with high potential energies could be enhanced to ensure the generalization ability of the models.”

14. It would also be beneficial to provide additional clarification in the text, particularly in Section 2.1, to explicitly explain the aspects of ViSNet's approach that contribute to its improved performance.

Response: Thank you for your valuable suggestion. We agree with your comment and would like to clarify that the performance improvement of ViSNet primarily stems from the following aspects:

1. The introduction of dihedral torsion and improper terms in the RGC module: This allows for the extraction of additional information, which significantly contributes to the improved performance.
2. Careful design in the ViS-MP module: This design ensures that the extracted information from the RGC module can be effectively integrated to obtain meaningful geometric representations.

3. Incorporation of higher-order spherical harmonics. This claim is further supported by Supplementary Table S3.

In response to your comment, we have revised Section 2.1 to include a detailed discussion of these factors. Thank you.

15. “It is worth noting that ViSNet is an energy-conserving potential, i.e., the predicted atomic forces are derived from the negative gradients of the potential energy with respect to the coordinates [23].” [23] Chmiela, S., Sauceda, H. E., Muller, K.-R. & Tkatchenko, A. Towards exact molecular dynamics simulations with machine-learned force fields. Nature communications 9, 1–10 (2018).

Citation is missing to Ref. 28.

Response: We have included the relevant citation in the corresponding statement. Thank you.

16. “In addition, considering that angle and dihedral are important potential terms in empirical force fields, the interpretability of the operations in the RGC strategy provides some insights in constructing hybrid force fields by combining empirical terms with deep learning.”

I do not understand how the presented interpretability can be used, if at all.

Response: We clarify that the design of RGC module in ViSNet originates from classical force fields as shown in Fig. 2 of the manuscript, while also breaking through the constraints of chemical bonds, thus enabling the acquisition of highly precise force fields through deep learning. To avoid potential misunderstandings, we removed the sentence in the revised manuscript.

17. “triplet and quadruplet interactions” are confusing terms.

Response: “triplet interactions” denotes angle information constituted by three atoms, whereas “quadruplet interactions” denotes the dihedral torsion or improper information constituted by four atoms. We have explained the two terms in the revised manuscript. Thank you.

Response to Reviewer #2

In the past decade, with the rapid development of deep learning, revolutionary breakthroughs were achieved in the fields of machine learning and artificial intelligence. More recently, it has permeated into various fields of science, among which an important topic is the machine-learning-based force field. In the submitted manuscript, the authors reported an exciting progress in this direction. More specifically, they propose a runtime geometry calculation (RGC) to effectively encode the geometric (conformational) information of molecules and Vector-Scalar interactive graph neural Network (ViSNet) as a machine learning model for molecular force field. The ViSNet shows remarkable performances in a few tested cases. For example, it achieved the top winners of PCQM4Mv2 track in the OGB-LCS@NeurIPS2022 competition, and it outperforms the state-of-the-art approaches on the molecules in the MD17 dataset, with only 0.7% samples (i.e., 950 samples for model training). This is an exciting achievement since deep learning usually requires a lot of data. It may have a profound impact in the field of machine-learning-based force field. I recommend the manuscript to be published after proper revision:

Response: We are grateful for Reviewer #2's time, effort, and valuable feedback. By revising the manuscript according to your nice suggestions, we believe the quality of the manuscript has been substantially improved. Thank you.

(1) According to the usual practice, Microsoft Research should be the first affiliation.

Response: We have updated the first affiliation in the revised manuscript. Thank you.

(2) p6: "MD17 consists of the MD trajectories of 7 small organic molecules": 17 but not 7 small organic molecules, or 7 small organic molecules and 10 other molecules?

Response: The MD17 dataset was introduced in 2017 by Chmiela et al. through their publication titled "Machine learning of accurate energy-conserving molecular force fields" (Sci. Adv. 2017;3(5): e1603015). This dataset offers molecular dynamics (MD) trajectories for seven small organic molecules, contributing valuable information to the field.

(3) Eq.(3): the dot "." should be deleted to avoid possible misunderstanding that it is an inner product.

Response: We have deleted the dot "." in the Eq. (3). Thank you.

(4) Chignolin is an (artificial) mini-protein. It is not appropriate to call it protein.

Response: Fixed. Thank you.

Reviewer #3 (Remarks to the Author):

Summary:

The paper introduces a graph neural network that incorporates a new geometric feature extraction module that models multi-body interaction in atom graphs with reduced computational costs. The proposed model achieves state-of-the-art performance on several molecular dynamics and property prediction tasks.

Geometric information such as angles, dihedral torsions, and improper angles are essential for learning a good molecular representation of downstream tasks like fitting potential energy surfaces. However, they are usually expensive to model. This paper proposes a framework called Runtime Geometry Calculation that is capable of extracting the above geometric information with better efficiency compared to several existing models. The authors also validate the efficacy of the learned embedding by doing a clustering. While the proposed model is a worthy addition to the existing GNN architectures for molecular machine learning, there are several points in the current presentation that can be further improved and clarified.

Comment:

The Scalar2Vec and Vec2Scalar modules introduced in the Method section of this paper are in fact closely related to PaiNN and TorchMD-NET. The current description makes the contribution of this work entangled with these existing works, for example, attention is used in both this work and TorchMD-NET. The authors can elaborate more on the difference between their models and existing architectures, and potentially provide more motivations for the architectural design.

Response: We appreciate Reviewer #3's suggestions. It is true that both PaiNN and TorchMD-NET (also known as "ET" and used in the manuscript) use vectorized representation; however, they only employ the vectorized representation to calculate angle information. We extended this by elegantly computing 4-body interactions, i.e., dihedrals and improper angles.

The attention mechanism is a common architecture in modern neural networks, and both our ViSNet and ET employ it. The most significant difference between our model and ET is that in our model, the edge hidden representation employed in the attention mechanism is updated in each layer while the edge feature in ET is fixed without updates through the whole neural network. This greatly enhances ViSNet's ability to characterize geometric structures. Furthermore, we replaced the original coordinates with higher-order spherical harmonics to create a stronger version of ViSNet. We have revised our article to provide more insights for the architectural design for our readers as follows:

"Notably, beyond employing angular information that has been used in PaiNN [1] and ET [2], ViSNet further considers the dihedral torsion and improper angle calculation with higher geometric tensors."

“It is important to note that previous studies [1, 2] primarily focused on updating node features, whereas our approach updates both node and edge features during message passing, leading to a more comprehensive geometric representation.”

[1] Schütt, Kristof, Oliver Unke, and Michael Gastegger. "Equivariant message passing for the prediction of tensorial properties and molecular spectra." *International Conference on Machine Learning*. PMLR, 2021.

[2] Thölke, Philipp, and Gianni De Fabritiis. "Equivariant transformers for neural network based molecular potentials." *International Conference on Learning Representations*. 2021.

Under Table 2, the authors state that “ViSNet can achieve better results with longer convergence time.” Does it mean ViSNet converges slower than the other models? If so, does this imply ViSNet generally requires more training iterations than other models, especially compared with TorchMD-NET, which has a similar message passing protocol but without the RGC module

Response: We clarify that we trained ViSNet model with 3,000 epochs on rMD17 dataset as the performance was already quite satisfactory, while the early stopping condition was not triggered, indicating that the model could theoretically achieve even lower metrics. In practice, ViSNet convergence requires a comparable number of epochs to ET on the MD17 dataset, and for some larger molecules (such as aspirin), ViSNet needs relatively fewer epochs. Thus, the RGC module does not impede the ViSNet convergence rate. We have removed this footnote from Table 2 in the manuscript and included detailed convergence criteria in our revised manuscript as follows:

“Training is stopped if a maximum number of epochs is reached, or the validation loss does not improve for a maximum number of early stopping patience.”

More details about the hyperparameters of ViSNet can be found in Supplementary Table S4.

Following up the last question, how is the computational cost of ViSNet compared to PaiNN/TorchMD-NET?

Response: We have included a comparison of computational cost for ViSNet with other models, including NequIP, Allegro, GemNet-OC, ET and PaiNN evaluated on the 166-atom Chignolin dataset which was also demonstrated in the response to the sixth question of reviewer #1. More specifically, as shown in Fig. 6 in the response letter, although ViSNet is marginally slower than ET and PaiNN, it introduces more geometric information, significantly enhancing its performance. Furthermore, when compared to GemNet, which also incorporates dihedral angles, ViSNet's computational cost is significantly more affordable. Similarly, ViSNet proves to be computationally efficient when compared to models employing the CG-product method, such as Allegro and NequIP.” We have added the experiments in Fig. 6 in the response letter and some analysis in the revised manuscript. Thank you.

Fig. 6. The comparison of model performance (Y-axis), training time consumption (X-axis), and model size (volume) among ViSNet (red), PaiNN, ET, GemNet-OC, Allegro, and NequIP (grey) evaluated on the Chignolin dataset.

Reviewer #1 (Remarks to the Author):

The authors have made significant improvements to their earlier version by performing the dynamics of Ac-Ala3-NHMe, conducting a thorough comparison with other models on Chignolin, and providing more clarity regarding the novelty of the proposed framework. However, there are still a few points that require further attention. Detailed comments:

It is puzzling to see that the dynamics of the tetrapeptide in VisNet and sGDML are nearly identical, despite a notable difference in errors that favors VisNet. Could authors elaborate on that? It would also be valuable to know the duration of the simulation. Given that there are three different phi/psi angles in the tetrapeptide, could the authors specify which one they plotted in their analysis?

"To the best of our knowledge, we are the first to implement four-body interactions, particularly dihedral torsion angles, with linear complexity, compared to GemNet. Other work in the ACE series can only describe 4-body interactions within one cluster, and their work does not encompass dihedral angles."

While the inclusion of a 4-body term is certainly a valuable addition, there is no discussion of MACE, which is many-body within the cutoff. In particular, it would be highly beneficial to incorporate an assessment of MACE performance into Fig. 6a and Fig. 7.

"VisNet achieved the lower MAE and the higher R2 score."

In Fig. S4, it would be helpful to supplement the MAE and R2 metrics with graphical representations such as kernel density estimation or 'violin' plots. These visualizations can offer a more comprehensive depiction of the error distribution, providing valuable insights that go beyond a single MAE measure, as exemplified in Fig. 3 of the referenced work (doi.org/10.1063/5.0139611).

Does the separate examination of each force component in Fig. 6c provide additional information? It might be more informative to analyze them collectively. The same applies to Fig. S5.

Other minor comments:

In Fig. 5, please clarify what the bar represents (0 to 7).

The interpretation of the naming "D3 backbone" is unclear (also G7, E6, T8). Visual illustration would be easier to understand.

On page 11, the word "furthermore" appears three times in a relatively small portion of the text. Some rephrasing may be needed.

Regarding Table 3, it's worth considering whether reporting 4-5 significant digits is meaningful or physically relevant. The same consideration applies to percentages such as 33.6%, 6.51%, 6.1%, and 4.1%. It might be more appropriate to report three significant digits and round percentages to integer values for clarity.

Reviewer #2 (Remarks to the Author):

The manuscript has been properly revised, and can now be accepted for publication.

Reviewer #3 (Remarks to the Author):

The authors responded to all my questions, thanks.

Reviewer #1 (Remarks to the Author):

The authors have made significant improvements to their earlier version by performing the dynamics of Ac-Ala3-NHMe, conducting a thorough comparison with other models on Chignolin, and providing more clarity regarding the novelty of the proposed framework. However, there are still a few points that require further attention. Detailed comments:

Answer: Thank you for taking the time to carefully review our manuscript once more. We highly value your expertise, and your critical feedback is instrumental in enhancing the quality of our work. Your commendations on our improvements are sincerely appreciated, and we are dedicated to addressing your remaining concerns. Considering your feedback, we have revisited the manuscript and made additional experiments and adjustments that we believe satisfactorily solve the new comments. Thanks for your professional input.

It is puzzling to see that the dynamics of the tetrapeptide in VisNet and sGDML are nearly identical, despite a notable difference in errors that favors VisNet. Could authors elaborate on that? It would also be valuable to know the duration of the simulation. Given that there are three different phi/psi angles in the tetrapeptide, could the authors specify which one they plotted in their analysis?

Answer:

The total simulation time for ethanol is 500 ps with a time step of 0.5 fs driven by DFT, SGDML and ViSNet, respectively. The total simulation time for Ac-Ala3-NHMe is 200 ps with a time step of 1 fs driven by DFT, SGDML and ViSNet, respectively.

To analyze the Ramachandran plot of different simulations, the free energy value was estimated using the potential of mean force (PMF). Phi and psi were set as two reaction coordinates (x, y). All three phi and psi dihedrals in Ac-Ala3-NHMe were calculated and plotted. The relative free energy value was calculated referred with the minimum value. To generate the landscape, 40 bins were used in both the x and y directions.

In addition, even though ViSNet showed better performance than sGDML for various conformations in the MD22 dataset, starting from the same structure of the alanine tetrapeptide, the performance difference may not have notable impact to the sampling efficiency for such small molecules, and thus may also led to similar dynamics of the tetrapeptide as shown in the Fig.5 of the manuscript.

We have added the descriptions and discussion in the revised manuscript. Thank you.

“To the best of our knowledge, we are the first to implement four-body interactions, particularly dihedral torsion angles, with linear complexity, compared to GemNet. Other work in the ACE series can only describe 4-body interactions within one cluster, and their work does not encompass dihedral angles.”

While the inclusion of a 4-body term is certainly a valuable addition, there is no discussion of MACE, which is many-body within the cutoff. In particular, it would be highly beneficial to incorporate an assessment of MACE performance into Fig. 6a and Fig. 7.

Answer: Per your suggestion, we trained MACE on the Chignolin dataset with default settings. As shown in Fig.6b of the revised manuscript (Fig. R1b in the response letter), MACE made larger errors for the six representative structures compared with other MLFFs. We then evaluated the energy and force prediction results for all conformations on the test sets. As shown in Fig. S4 & S6 in the supplementary material (Fig. R2 in the response letter), MACE exhibited worse energy prediction results than other MLFFs but the second-best force prediction results.

Fig. R1. Applications of ViSNet for Chignolin conformational space evaluation and MD simulations. (a) The visualization of Chignolin structure. The backbone is colored grey while the side chains of each residue in Chignolin

are highlighted with ball and stick. (b) The energy landscape of Chignolin sampled by REMD. The x-axis of the landscape is the distance between carbonyl oxygen on D3 backbone and nitrogen on G7 backbone, while the y-axis is the distance between carbonyl oxygen on E5 backbone and nitrogen on T8 backbone. 6 structures were then selected for visualization. Each structure is shown as cartoon and residues are depicted in sticks. The histograms show the absolute error between the energy difference predicted by MLFFs including ViSNet, ET, PaiNN, GemNet-OC, NequIP, Allegro and MACE or calculated by MM, and the ground truth calculated by DFT on the corresponding structure.

Fig. R2. The energy (a) and force (c-d) correlations between the ground truth calculated by DFT and predictions by MACE. The corresponding distributions of energy and force predictions are also shown in each panel.

We further made an evaluation on the model performance, training time consumption, and the model size of MACE on Chignolin dataset and compared with others. As shown in Fig. 7 in the revised manuscript (Fig. R3 in the response letter), ViSNet is faster and smaller than MACE thanks to streamlining the CG-product and achieves better performance than MACE due to the novel designed Runtime Geometric Calculation and Vector-Scalar interactive message passing.

Fig. R3. The comparison of model performance (Y-axis), training time consumption (X-axis), and training memory consumption (volume) among ViSNet (red) and other algorithms (grey) including PaiNN, ET, MACE, GemNet-OC, Allegro and NequIP on Chignolin.

We have redesigned Fig.6, Fig.7, Fig. S4 and Fig. S6 in the revised manuscript and added corresponding descriptions in the revised manuscript. Thank you.

In Fig. S4, it would be helpful to supplement the MAE and R2 metrics with graphical representations such as kernel density estimation or 'violin' plots. These visualizations can offer a more comprehensive depiction of the error distribution, providing valuable insights that go beyond a single MAE measure, as exemplified in Fig. 3 of the referenced work (doi.org/10.1063/5.0139611).

Answer: Per your suggestion, we have made violin plot of the absolute error between the ground truth calculated by DFT and predictions or calculations by MLFFs and molecular mechanics (MM) respectively on the test dataset in Fig. S5 of Supplementary Material (Fig. R4 in the response letter). As shown in the figure, ViSNet, PaiNN and ET exhibited smaller errors than other MLFFs while

MM got a much wider range of prediction errors. We have added descriptions in the revised manuscript.

Fig. R4. The violin plot of absolute errors between the ground truth calculated by DFT and predictions or calculations by MLFFs and molecular mechanics (MM) respectively on the test dataset.

Does the separate examination of each force component in Fig. 6c provide additional information? It might be more informative to analyze them collectively. The same applies to Fig. S5.

Answer: From the analysis of each force component, we found ViSNet as well as other MLFFs has consistent performance among each force component, which implies the machine learning force fields don't have bias towards any force component. We have added some descriptions in the revised manuscript. Thanks.

Other minor comments:

In Fig. 5, please clarify what the bar represents (0 to 7).

Answer: The bar represents the values of free energy. We have added descriptions in the revised manuscript.

The interpretation of the naming “D3 backbone” is unclear (also G7, E6, T8). Visual illustration would be easier to understand.

Answer: We have added Fig. 6a in the revised manuscript (Fig. R1a in the response letter) for better visual illustration. Thank you.

On page 11, the word "furthermore" appears three times in a relatively small portion of the text. Some rephrasing may be needed.

Answer: We have revised the descriptions in the revised manuscript. Thank you.

Regarding Table 3, it's worth considering whether reporting 4-5 significant digits is meaningful or physically relevant. The same consideration applies to percentages such as 33.6%, 6.51%, 6.1%, and 4.1%. It might be more appropriate to report three significant digits and round percentages to integer values for clarity.

Answer: Fixed. Thank you.

Reviewer #1 (Remarks to the Author):

The authors have successfully addressed my comments on the manuscript.